**Brief Communication**

# SpatialData: an open and universal data framework for spatial omics

Luca Marconato[1,2,3,19], Giovanni Palla[4,5,19], Kevin A. Yamauchi [6,7,19], Isaac Virshup[4,19], Elyas Heidari[1,2,8], Tim Treis [2,4], Wouter-Michiel Vierdag [1], Marcella Toth[4], Sonja Stockhaus [4,9], Rahul B. Shrestha[4], Benjamin Rombaut [10,11,12], Lotte Pollaris [10,11,12], Laurens Lehner[4,9], Harald Vöhringer[1,13,14], Ilia Kats [2], Yvan Saeys [10,11,12], Sinem K. Saka [1], Wolfgang Huber [1], Moritz Gerstung[8], Josh Moore [15,16] ✉, Fabian J. Theis [4,5,17,18] ✉ & Oliver Stegle [1,2,18] ✉

Spatially resolved omics technologies are transforming our understanding of biological tissues. However, the handling of uni- and multimodal spatial omics datasets remains a challenge owing to large data volumes, heterogeneity of data types and the lack of flexible, spatially aware data structures. Here we introduce SpatialData, a framework that establishes a unified and extensible multiplatform file-format, lazy representation of larger-than-memory data, transformations and alignment to common coordinate systems. SpatialData facilitates spatial annotations and cross-modal aggregation and analysis, the utility of which is illustrated in the context of multiple vignettes, including integrative analysis on a multimodal Xenium and Visium breast cancer study.

The function of biological tissues is strongly linked to their composition and organization. Advances in imaging and spatial molecular profiling technologies enable the addressing of these questions by interrogating tissue architectures with ever-growing comprehensiveness, resolution and sensitivity[1,2]. Existing spatial molecular profiling methods quantify DNA, RNA, protein and/or metabolite abundances in situ[3,4]. Several of these technologies employ light microscopy, providing spatial resolution of morphological features at length scales from the subcellular to entire organisms. Spatial omics technologies are advancing rapidly, and individual data modalities and methods feature distinct advantages

and limitations such as trade-offs in spatial resolution, the extent of molecular multiplexing and detection sensitivity. The ability to efficiently integrate and then operate on data from different spatial omics modalities promises to be instrumental for the construction of holistic views of biological systems.

While progress has been made in the analysis of individual spatial omics datasets, integration of uni- and multimodal spatial omics data entails important practical challenges not sufficiently addressed by existing solutions[5–7] (Extended Data Table 1, Supplementary Note 1 and Supplementary Table 1). Even basic operations such as loading of

[1]European Molecular Biology Laboratory, Genome Biology Unit, Heidelberg, Germany. [2]Division of Computational Genomics and System Genetics, German Cancer Research Center, Heidelberg, Germany. [3]Collaboration for joint PhD degree between EMBL and Heidelberg University, Faculty of Biosciences, Heidelberg, Germany. [4]Institute of Computational Biology, Helmholtz, Center Munich, Munich, Germany. [5]TUM School of Life Sciences Weihenstephan, Technical University of Munich, Munich, Germany. [6]Department of Biosystems, Science and Engineering, ETH Zürich, Basel, Switzerland. [7]Swiss Institute of Bioinformatics, Basel, Switzerland. [8]Division of Artificial Intelligence in Oncology, German Cancer Research Center, Heidelberg, Germany. [9]TUM School of Computation, Information and Technology, Technical University of Munich, Munich, Germany. [10]Data Mining and Modeling for Biomedicine, VIB Center for Inflammation Research, Ghent, Belgium. [11]Department of Applied Mathematics, Computer Science and Statistics, Ghent University, Ghent, Belgium. [12]VIB Center for AI and Computational Biology, Ghent, Belgium. [13]Molecular Medicine Partnership Unit, Heidelberg, Germany. [14]Department of Medicine V, Hematology, Oncology, and Rheumatology, University of Heidelberg, Heidelberg, Germany. [15]German BioImaging – Gesellschaft für Mikroskopie und Bildanalyse e.V, Konstanz, Germany. [16]Open Microscopy Environment Consortium, Munich, Germany. [17]Department of Mathematics, Technical University of Munich, Munich, Germany. [18]Cellular Genetics Programme, Wellcome Sanger Institute, Cambridge, UK. [19]These authors contributed equally: Luca Marconato, Giovanni Palla, Kevin A. Yamauchi, Isaac Virshup. ✉e-mail: josh@openmicroscopy.org; fabian.theis@helmholtz-munich.de; oliver.stegle@embl.de

datasets into analysis pipelines in a coherent manner is hampered by the diversity in data types (for example, tabular data for sequencing and tens- to hundreds-of-gigabyte dense arrays for images) and file formats (for example, technology-specific vendor formats). In addition, individual spatial omics modalities can differ vastly in spatial resolution and the spatial regions for data acquisition in a tissue are often not aligned. Thus, for integration of such data they must be appropriately transformed and aligned to a common coordinate system (CCS), which is a prerequisite for the establishment of global common coordinate frameworks (CCFs)[8]. Finally, untangling the complexity of multimodal spatial omics datasets requires expert knowledge and motivation of approaches that enable large-scale interactive data exploration and annotation. Thus, to unlock the full potential of emerging spatial multiomics studies[2,9] there is a need for computational infrastructures to store, explore, analyze and annotate data across the full breadth of spatial omics technologies with a unified programmatic interface.

The SpatialData framework enables the findable, accessible, interoperable, reusable (FAIR)[10] integration of multimodal spatial omics data. A language-independent storage format increases the interoperability of data sources while the Python library standardizes access of, and operation across, different data types. The SpatialData format supports all major spatial omics technologies and derived quantities (Fig. 1a,c, Supplementary Note 2 and Supplementary Table 2). Briefly, spatial datasets are represented using five primitive elements: Images (raster images), Labels (for example, raster segmentation masks), Points (for example, molecular probes), Shapes (for example, polygon regions of interests, array capture locations and so on) and Tables (for example, molecular quantifications and annotations) (Supplementary Tables 2 and 3). The file format also tracks coordinate transformation or alignment steps applied to individual datasets. Dataset collections can be stored within a single SpatialData store, thereby facilitating joint integrative analyses. The SpatialData format builds on the Open Microscopy Environment–Next-Generation File Format (OME–NGFF) specifications and leverages the Zarr file format (Supplementary Fig. 1), thereby offering performant, interoperable access for both traditional file system- and cloud-based storage[11,12] (Supplementary Note 3).

The SpatialData Python library represents this format as SpatialData objects in memory, which supports lazy loading of larger-than-memory data (Fig. 1b). The library also provides reader functions for widely used spatial omics technologies (Fig. 1c and Supplementary Table 3), as well as versatile functionalities for manipulating and accessing SpatialData objects and to define CCSs of biological tissues[8]. Briefly, each individual dataset is associated with a modality-specific coordinate transformation (Fig. 1b) that includes affine transformations and composite operations. Once aligned, a collection of datasets can be queried (Extended Data Fig. 1) and aggregated (Extended Data Fig. 2)—for example, using spatial annotations at diverse scales (cells, grids, anatomical regions) and both within and across modalities. The query and aggregation interfaces also allow for the creation of new datasets grouped by biologically informed factors from large dataset collections, thereby facilitating exploration, selected data sharing and access.

SpatialData has a napari plugin for interactive annotation (napari-spatialdata; Fig. 1d and Extended Data Fig. 3). The napari-spatialdata plugin can be used for the interactive definition of spatial annotations such as drawing regions of interest, or to define landmarks for guiding multidataset registration. Static figures and graphics can be created using the spatialdata-plot library (Extended Data Fig. 4).

The SpatialData library integrates seamlessly with the Python ecosystem by building on standard scientific Python data types. We have implemented a PyTorch Dataset class to effectively train deep learning models directly from SpatialData objects (Fig. 1e, Supplementary Note 4 and Extended Data Fig. 5). Further, thanks to the modular nature of the data representation, analysis packages in the scverse[13] ecosystem

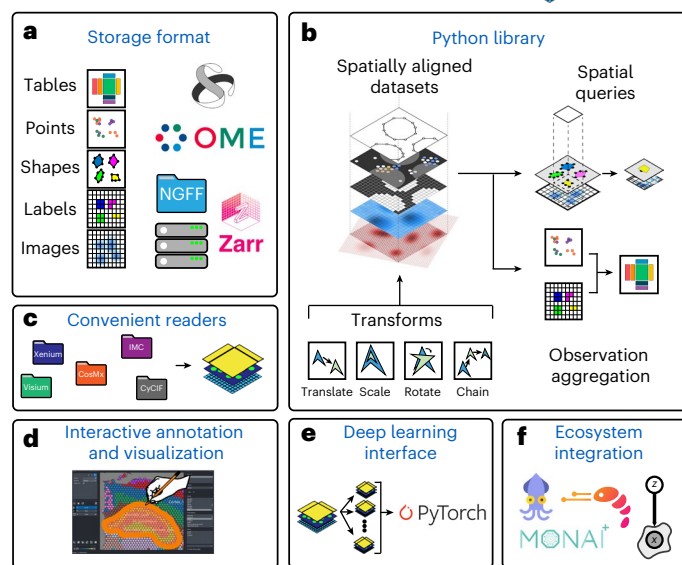

**Fig. 1 | Design overview and core functionality of SpatialData. a**, The SpatialData storage format represents raw and derived data from a wide range of spatial omics technologies in a unified manner. The format builds on five primitive elements (SpatialElements), which are serialized to a Zarr store in an OME–NGFF-compliant manner. **b**, The SpatialData Python library implements operations for data access, alignment, queries and aggregation of spatial datasets. Coordinate transformations can be specified to align multiple modalities to a CCS, allowing for deployment of spatial queries and aggregation operators across modalities. **c**, SpatialData is compatible with common data formats, including vendor-specific file formats. Collections of datasets can be stored in a single Zarr store and are represented as a SpatialData object. **d**, Datasets stored in SpatialData format can be annotated interactively using the integrated napari-spatialdata plugin; SpatialData provides functionality for the generation of both interactive and static plots. **e**, SpatialData implements a PyTorch Dataset class, thereby facilitating the training of deep learning models directly from SpatialData objects. **f**, SpatialData builds on established standards and software, thereby providing interoperability with existing multimodal analysis approaches including Squidpy[15], Scanpy[14], MONAI[23] and scvi-tools[24], among others.

such as Scanpy[14], Squidpy[15] and scvi-tools[16] can be used for analysis of SpatialData objects (Fig. 1f and Supplementary Fig. 2). Taken together, the SpatialData framework provides infrastructure for the integration and analysis of spatial omics data.

To illustrate the utility of SpatialData for multimodal integration and analysis, we used the framework to represent and process data from a breast cancer study that combines hematoxylin and eosin (H&E) images and 10x Genomics Visium and Xenium assays[17]. The study comprises two in situ sequencing datasets (Xenium) and one spatial transcriptomics dataset (10x Visium CytAssist) from consecutive sections of a breast cancer tumor. First we used napari-spatialdata to define landmark points present in all datasets, followed by alignment of all three datasets using transformations to define a CCS (Fig. 2a). As a result of the alignment, SpatialData enabled us to identify the common spatial area, which can be accessed using SpatialData queries across datasets.

Next we used the collective information from all three datasets to create a shared set of spatial annotations. Briefly, we selected four regions of interest (ROIs) based on histological features present in the H&E image using napari-spatialdata (Extended Data Fig. 6). We then used genome-wide transcriptome information in Visium to estimate copy number states (using CopyKat[18]) and to annotate major genetic subclones. Finally we annotated cell types in two Xenium replicates by transferring cell-type labels from an independent breast cancer

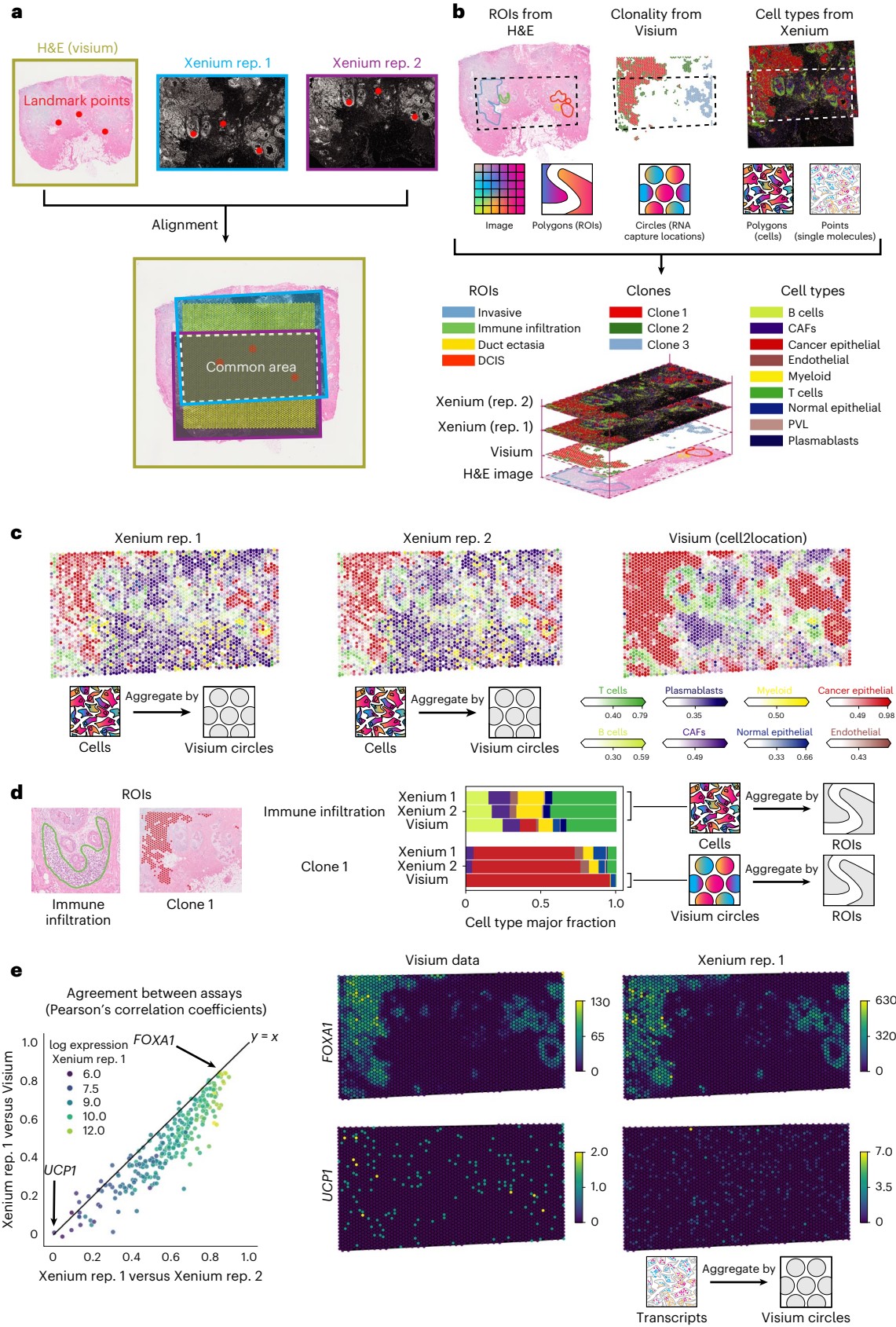

single-cell RNA sequencing (scRNA-seq) atlas[19] (ingest, implemented in scanpy[14]; Fig. 2b).

To exemplify how SpatialData can be used to transfer spatial annotations between datasets, we considered the masks from Visium capture locations and aggregated cell-type information from the overlapping

Xenium cells to estimate cell-type fractions at each location. For comparison we also considered a deconvolution-based analysis of Visium counts (using cell2location[20]) with the same scRNA-seq-derived cell types[19] as reference. We observed high concordance of cell-type abundance estimates between Xenium replicates (median Pearson's $R$ = 0.88

**Fig. 2 | Alignment and integrative analysis of three spatial datasets from breast cancer. a**, Registration of two breast cancer Xenium replicate (rep.) slides, one Visium slide and their corresponding H&E images to a CCS based on interactively selected landmarks. **b**, Illustration of how spatial annotations can be transferred across datasets using the CCS. From top to bottom, spatial annotations derived from multiple datasets, including histological regions (H&E image), tumor clones (Visium-derived copy number aberrations) and cell types (Xenium and scRNA-seq). Spatial annotations, represented by different spatial elements (polygons, circles, molecules), can be transferred between datasets via the CCS. **c**, SpatialData queries facilitate cross-modality aggregation, quality control and benchmarking. Left and middle, cell-type fractions in Xenium computed at circular regions corresponding to Visium quantification locations; right, cell-type fraction estimates from deconvolution methods based

on Visium data (using cell2location). **d**, Use of SpatialData queries for arbitrary geometrical quantifications. Shown are cell-type fraction estimates obtained in Xenium (derived from the paired scRNA-seq dataset) and Visium (cell2location estimates) at annotated ROIs and clones as in **b**. **e**, Comparison of gene expression quantification in Xenium and Visium using SpatialData aggregations at Visium capture locations. Left, scatter plot of the correlation coefficient of aggregated gene expression quantifications between Xenium replicates (*x* axis) versus that between Xenium and Visium (*y* axis). Shown are gene expression quantifications for 313 genes (dots) present in both Xenium and Visium. Color denotes log expression in Xenium replicate 1. Right, visualization of aggregated expression levels at Visium locations for *FOXA1* (top) and *UCP1* (bottom). Color bars denote raw counts.

across Visium locations) and overall good agreement between Xenium- and deconvolution-based estimates (median Pearson's $R = 0.69$).

Analogous to the aggregation at Visium locations, we considered ROIs defined from H&E and areas defined by the union of subclone locations from Visium (Fig. 2d and Supplementary Fig. 3a). Again we quantified cell-type fractions within each region, either directly using cell count fractions from Xenium or via deconvolution of the corresponding Visium data. The two Xenium replicates showed high concordance of cell-type fractions, and Xenium and Visium were consistent.

As a second aggregation use case we compared expression estimates for individual genes at Visium capture locations using either Xenium or Visium data. We again transferred Visium capture locations to aggregate individual molecule counts from Xenium into the Visium masks (Fig. 2e and Supplementary Fig. 3b). As expected, the aggregated counts were highly concordant between Xenium replicates (median Pearson's $R = 0.62$; Fig. 2e and Supplementary Fig. 3c–e) and, to a lesser extent, between Xenium and Visium counts (median Pearson's $R = 0.48$; Supplementary Fig. 3c–e). We also noted a direct relationship between overall transcript abundance and the agreement between different tissue sections and technologies (Fig. 2e).

In sum, these examples illustrate the flexibility of the aggregation functionality that can be applied between SpatialElements of different kinds (points, circular capture locations, cells and larger anatomical ROIs) to transfer diverse types of spatial annotation (cell expression, cell-type fractions). Further examples and advanced-use cases of SpatialData aggregation operations are discussed in Extended Data Fig. 2.

SpatialData facilitates the processing of a wide range of uni- and multimodal datasets. The online documentation of SpatialData comes with vignettes that illustrate additional use cases. For example, we illustrate how SpatialData can serve as a backend to facilitate the training of deep learning models (Extended Data Fig. 5 and Supplementary Note 4), and to conduct downstream analysis using spatial interpretation tools such as Squidpy (Supplementary Fig. 2). As a starting point for using SpatialData in conjunction with different technologies, we also currently provide preformatted SpatialData objects from >40 datasets acquired by eight technologies (Supplementary Table 2). Interactive annotation can be performed on both single- and multimodality datasets. Finally we explored how SpatialData can align multiple fields of view into a global reference coordinate system by mapping 12 Visium slides to a large prostate section (Extended Data Fig. 7). Further information, including comprehensive documentation of the SpatialData Python library, tutorials, example datasets and a contributor guide, is available online (https://spatialdata.scverse.org).

Here we present SpatialData, a flexible, community standards-based framework for storage, processing and annotation of data from virtually any spatial omics technology available to date. The ability to flexibly create common coordinate systems by aligning datasets is a critical cornerstone to establishing comprehensive CCFs, which will unlock new analysis approaches that facilitate robust comparison and reuse of samples across studies. In conclusion, the flexibility and readily accessible solutions provided by the SpatialData framework

enable new possibilities in analysis and enhance the reproducibility of integrated spatial analysis.

As the uptake of SpatialData continues to grow its utility will increase further. Ongoing developments (discussed in Supplementary Notes 5 and 6) extend the interoperability of SpatialData with R/Bioconductor[21], provide support for multiscale point and polygon representations—such as polygonal meshes and five-dimensional volumetric images (that is, *czyx* images with an additional time component)—and support cloud-based data access both programmatically and via the visualization tool Vitessce[22]. In summary, SpatialData provides an open and universal data framework for spatial omics.

## Online content

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

## Methods

### SpatialData framework
The SpatialData framework comprises a core package, spatial data and associated satellite packages napari-spatialdata, spatialdata-io and spatialdata-plot, compatible with Python 3.9 and above. All code is available on GitHub as part of the scverse organization and is licensed under the permissive 'BSD 3-Clause License'. The project structures inherit from the scverse cookiecutter and the napari plugin cookiecutter, thus implementing unit tests and precommit checks in a continuous integration setting. The documentation is built using Sphinx and hosted on Read the Docs. It includes application programming interface (API) descriptions, example notebooks and a table with links to downloadable spatial omics datasets. Each dataset can be downloaded in full (.zip) or even directly accessed from the cloud (public S3 storage). Documentation, tutorials and sample data can be found in the links below.

- Documentation: https://spatialdata.scverse.org
- Installation instructions: https://spatialdata.scverse.org/en/latest/installation.html
- Tutorials: https://spatialdata.scverse.org/en/latest/tutorials/notebooks/notebooks.html
- Sample data: https://spatialdata.scverse.org/en/latest/tutorials/notebooks/datasets/README.html

We also provide a contribution guide and technical design document to encourage adoption. Users can reach out to the core development team via the GitHub Issues bug-tracking system. To encourage collaboration between the imaging and scverse communities we have created a public chat stream on the imagesc Zulip messaging platform: https://imagesc.zulipchat.com/#narrow/stream/329057-scverse.

### SpatialData framework dependencies
The framework depends on routinely used Python libraries. In detail, the spatialdata package depends on networkx, numpy (scientific stack), anndata (single-cell data), dask-image, multiscale-spatial-image, ome-zarr-py, spatial-image, xarray, xarray-schema, xarray-spatial, zarr (raster spatial data), geopandas, pyarrow, pygeos, shapely (vector spatial data), fsspec, rich, tqdm, typing_extensions (utilities) and torch (deep learning, optional dependency).

The satellite packages spatialdata-io, spatialdata-plot and napari-spatialdata require additional dependencies; we refer the reader to the Reporting Summary for a complete list, and to the pyproject.toml and setup.cfg files of the corresponding GitHub repositories for the most up-to-date list, as the packages and their dependency continuously evolve.

All packages in the SpatialData framework are routinely published to PyPI via GitHub Actions and, as such, pip can be used readily to install the software and all its dependent libraries. Conda support is in preparation.

### Raw human breast cancer Xenium and Visium data
We downloaded the raw data from https://www.10xgenomics.com/products/xenium-in-situ/preview-dataset-human-breast.

### Loading Xenium and Visium datasets into SpatialData
The 10x Xenium and Visium readers from spatialdata-io were used to read the data into SpatialData objects. For the Xenium datasets, the DAPI channel was stored as a multiscale Image, cell and nuclei segmentation masks and boundaries were stored as Shapes elements whereas the transcripts were stored as Points. The metadata and count matrices were stored as a Table in the SpatialData object. For the Visium dataset, the H&E image was stored as a multiscale Image, the array capture areas (circles) were stored as Shapes and the count matrix and annotations were stored in the Table.

### Cell-type annotation of Xenium replicates
We annotated cells from Xenium replicates using a publicly available scRNA-seq breast cancer atlas[19] comprising nine malignant and normal

cell types and 29 subtypes. After subsetting the atlas to the subset of 313 genes present in the Xenium panel, we applied the ingest method for label transfer as implemented in the Scanpy package (v.1.9)[14] to annotate cells from the Xenium replicates. We transferred major cell-type labels first (coarse grained) and then within each class we mapped minor cell types (fine grained). In the current analysis only major cell types are shown. The nine major cell types are B cells, cancer-associated fibroblasts (CAFs), cancer epithelial, endothelial, normal epithelial, plasmablasts and perivascular-like cells (PVL) and T cells.

### Alignment to create common coordinate systems
We selected three landmark points from the images from the two Xenium replicates and the Visium dataset. Landmark points are to be selected on each of the images in the same order and there should be a 1-to-1 spatial correspondence between sets of points. Xenium replicate 1 was used as the reference to which Xenium replicate 2 and Visium were aligned using the SpatialData function align_elements_using_landmarks. We used napari-spatialdata to annotate the landmark points and to view the result of alignments. Internally, Dask's lazy-loading and Zarr's multiscale representation made it possible to performantly explore and zoom the datasets, even in a low-memory device like a standard laptop.

### Computation of cell-type fractions for Visium
Following alignment, the shared area between each cell and from the Xenium replicates and Visium locations was computed. Cell-type fractions were then computed for each Visium location based on the surface fractions of the locations covered by each cell type. This was done using the SpatialData aggregate function with fractions=True, and was performed separately for Xenium replicates 1 and 2.

### Cell-type deconvolution using cell2location
We used cell2location (v.0.1.3)[20] to estimate cell-type fractions at Visium locations, with the aforementioned breast cancer atlas as the reference. For this task we operated on the subset of 313 genes present in the Xenium replicates and subset the Visium dataset and breast cancer atlas to those genes. We set the default parameters as suggested in the cell2location tutorial (https://cell2location.readthedocs.io/en/latest/notebooks/cell2location_tutorial.html). The analysis can be found at https://github.com/scverse/spatialdata-notebooks/tree/main/notebooks/paper_reproducibility. For visualization, only cell types contributing at least 5% per Visium capture location were taken into account then the quantity at each location was normalized to have a total of 1.

### ROI selection with napari-spatialdata
Following alignment, four ROIs were selected based on the H&E image from the Visium dataset using the napari-spatial data plugin, and these ROIs were then added to the aligned Xenium replicates. Each ROI was selected based on its distinct microanatomical characteristics and then labeled manually based on the underlying cell-type composition from the Xenium replicates.

### Clone detection on Visium using CopyKat
We used CopyKat (v.1.1.0)[18] with default parameters to estimate copy number states from the Visium count matrix followed by hierarchical clustering, which identified three major clusters on the locations labeled as 'aneuploid'; these three clusters were used as genetic subclones. We also transferred clone labels to overlapping cells from Xenium replicates; these labels were stored as a SpatialData table element. This analysis was conducted in R separately (the notebooks repository: https://github.com/scverse/spatialdata-notebooks/tree/main/notebooks/paper_reproducibility).

Visium's anndata table was saved in .h5ad AnnData format[14,25] for loading and analysis in R, and clone labels were then transferred back to SpatialData via .h5ad. There are ongoing efforts in the Bioconductor

community to enable direct loading of anndata tables into R from Zarr, such as anndataR[26], which would obviate the need for exporting as.h5ad (HDF5 format) when completed.

### ROI cell-type fractions

We next computed, for each ROI and clone, the fractions of cell types for the cells contained within them. The SpatialData aggregation APIs offer a convenient interface to compute these metrics, independently if what is being aggregated is a set of circles or polygons, and if the target region is a polygonal ROI or a set of circles defining a particular clone.

### Transcript aggregations

For each Visium capture location we aggregated transcripts from the Xenium replicates falling into each Visium location; we performed this analysis for Xenium replicates 1 and 2 separately. This yielded two aggregated count matrices that were saved as separate layers in Visium's SpatialData objects table.

### Reporting summary

Further information on research design is available in the Nature Portfolio Reporting Summary linked to this article.

## Data availability

We converted several example datasets to Zarr using the SpatialData package. At the time of writing we included data from the following technologies: NanoString CosMx, 10x Genomics Xenium, 10x Genomics Visium, CyCIF, MERFISH, MIBI-TOF and Imaging Mass Cytometry. The scripts used to convert data, as well as the converted data, are accessible from https://spatialdata.scverse.org/en/latest/tutorials/notebooks/datasets/README.html. For an overview of the datasets and their respective source publication please refer to Supplementary Table 2.

## Code availability

SpatialData is available as a Python package via pip, and comes with an extensive set of examples and tutorials that can be accessed from the documentation at https://spatialdata.scverse.org. Furthermore, the documentation also includes a contribution guide for researchers interested in participating in the design and implementation of the framework. All scripts used to reproduce the analyses included in this manuscript can be downloaded from the spatialdata-notebook repository: https://github.com/scverse/spatialdata-notebooks/tree/main/notebooks/paper_reproducibility.

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

## Acknowledgements

We thank the following individuals for their contributions: D. Bredikhin for participation in a hackathon in Basel (April 2022) focused on discussions on representations for multiple modalities and in the scverse ecosystem; B. Wadie, C. Tischer, S. Gonzalez Tirado and L. Hetzel for attending a hackathon in Heidelberg (June 2022); A. Lomakin for his contributions to discussions on alignment of clones and niches; O. Lazareva for contributing work on clonality for the breast cancer study during the de.NBI BioHackathon SpaceHack project in Lutherstadt-Wittenberg (December 2022); and organizers of and participants in the de.NBI BioHackathon SpaceHack project. We thank H. L. Crowell, C. Ahlmann-Eltze, M. Smith, N. Eiling, L. Meyer and L. Moses for valuable discussions on R interoperability and their prototype implementations of R readers for OME–Zarr and SpatialData objects. We also thank I. Gold, M. Keller, N. Gehlenborg, T. Li and O. Bayraktar for discussions and initial implementations concerning JavaScript interoperability for remote data visualization with Vitessce[22], in particular as part of the WebAtlas pipeline[27]. In addition we thank M. Klein for his valuable contributions to napari-spatialdata during a hackathon in Heidelberg (April 2023); F. Wünnemann for his contributions on spatialdata-io in another hackathon in Heidelberg (July 2023); Q. Blampey for contributions to spatialdata-io; A. Shmatko for contribution to implementation of the napari lasso tool; A. Defauw for his work on the apply function for raster data; M. McCormick for discussions and support regarding usage of the packages SpatialImage and MultiscaleSpatialImage; W. Moore for discussions on OME–NGFF and technical support on OME–Zarr; J. Bogovic for developing the OME–NGFF transformation specification; J. Lüthi and C. Mah for discussions during SpatialData meetings; A. S. Eisenbarth for bug fixes and general feedback; and T. Graf for his work on a prototype involving nonlinear transformations. In addition we thank group members of the Stegle and Theis laboratory for helpful discussions. Finally we acknowledge the respective funding programs of the authors. L.M. is supported by the EMBL International PhD Programme. G.P. is supported by the Helmholtz Association under the joint research school Munich School for Data Science and by the Joachim Herz Foundation. K.A.Y. was supported by the Open Research Data Program of the ETH Board and a Personalized Health and Related Technologies Transition Postdoc Fellowship (no. PHRT 2021–448). E.H. is supported by the DKFZ International PhD Programme. T.T. is supported by the Helmholtz Association under the joint research school Munich School for Data Science. W.-M.V. is supported by the EMBL International PhD Programme and by research funding from Cellzome, a GSK company. B.R. and Y.S. are supported by the Flanders AI Research Program. L.P. is supported by a PhD fellowship from The Research Foundation—Flanders (grant no. 11J7324N). Y.S. is supported by the FWO-EOS program and the BOF-GOA fund. H.V. Is supported by a BMBF grant (SIMONA). S.K.S. acknowledges core funding from the European Molecular Biology Laboratory and research funding from

Cellzome, a GSK company. J.M. was supported for work on OME–NGFF by grant nos. 2019-207272 and 2022-310144 and on Zarr by grant nos. 2019-207338 and 2021-237467 from Chan Zuckerberg Initiative DAF, an advised fund of Silicon Valley Community Foundation; and was funded by Deutsche Forschungsgemeinschaft (German Research Foundation, no. 501864659) as part of NFDI4BIOIMAGE. F.J.T. acknowledges support by the Helmholtz Association's Initiative and Networking Fund through Helmholtz AI (grant no. ZT-I-PF-5-01), by Wellcome Leap as part of the ΔTissue Program and by the Chan Zuckerberg Initiative DAF (advised fund of Silicon Valley Community Foundation, grant no. 2021-240328 (5022)). O.S. acknowledges support by Wellcome Leap as part of the ΔTissue Program. This project has been made possible in part by grant no. 2023- 323350 from the Chan Zuckerberg Initiative DAF, an advised fund of Silicon Valley Community Foundation.

## Author contributions

L.M., G.P., K.A.Y. and I.V. contributed equally. E.H., T.T., and W.V. contributed equally. L.M., G.P., K.A.Y. and I.V. designed and authored the spatialdata library, with contributions from I.K. during early prototyping. L.M., G.P., K.A.Y., I.V. and J.M. authored the spatialdata storage specification. G.P., L.M., M.T., W.V. and R.B.S. wrote napari-spatialdata. H.V. designed and prototyped spatialdata-plot with input from T.T., G.P. and L.M. T.T., G.P., S.K.S. and H.V. implemented spatialdata-plot. G.P., L.M., L.L. and W.-M.V. implemented the spatialdata-io library. E.H. performed analysis on the (Xenium and Visium) breast cancer dataset with input from L.M., G.P. and K.A.Y. E.H., W.-M.V., B.R. and L.P. contributed to library improvement. O.S., F.J.T. and J.M. supervised the work.

## Funding

## Competing interests

J.M. holds equity in Glencoe Software, which builds products based on OME–NGFF. F.J.T. consults for Immunai, Inc., Singularity Bio B.V., CytoReason Ltd, Cellarity and Omniscope and has ownership interest in Dermagnostix GmbH and Cellarity. O.S. is a paid consultant of Insitro, Inc., S.K.S. is a consulting scientific cofounder for Digital Biology, Inc. The remaining authors declare no competing interests.

## Additional information

**Extended data** is available for this paper at https://doi.org/10.1038/s41592-024-02212-x.

**Correspondence and requests for materials** should be addressed to Josh Moore, Fabian J. Theis or Oliver Stegle.

Polygon query

```python
from shapely import Polygon

polygon = Polygon(
    [
        (2000, 2000),
        (4500, 6000),
        (4500, 3500),
        (9000, 10000),
        (12000, 6000),
    ]
)
polygon
```

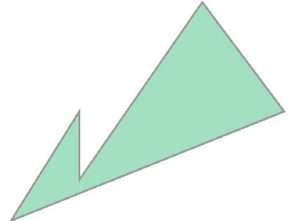

```python
from spatialdata import polygon_query

cropped_sdata2 = polygon_query(
    sdata=sdata_ST8059050,
    polygons=polygon,
    target_coordinate_system="ST8059050",
)

cropped_sdata2
```

Bounding box query

```python
cropped_sdata = sdata_ST8059050.query.bounding_box(
    axes=["x", "y"],
    min_coordinate=[8000, 12000],
    max_coordinate=[12000, 16000],
    target_coordinate_system="ST8059050",
)

cropped_sdata
```

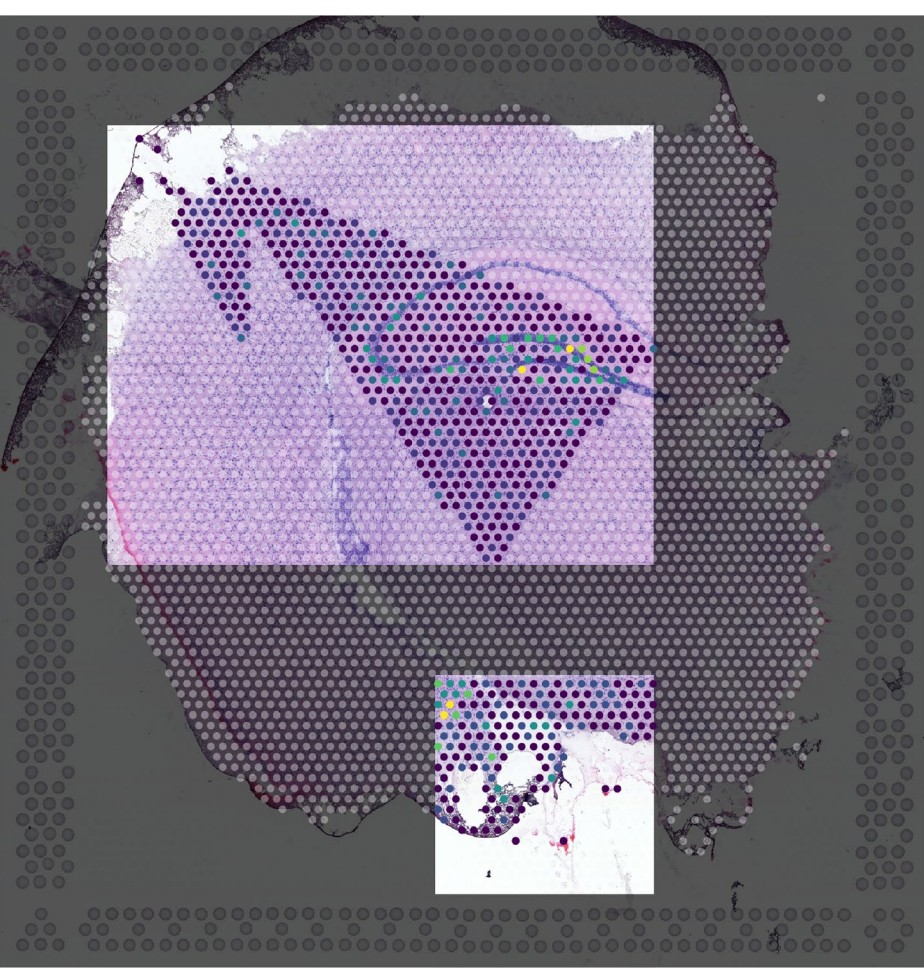

**Extended Data Fig. 1 | Illustration of the SpatialData query function.**
To facilitate analyses on large datasets, SpatialData enables the selection of distinct regions within a dataset. The spatial query interface allows users to request the data contained in a query region, which can be specified both as a bounding box or a polygonal region. The query region can be specified using any of the coordinate systems present in the SpatialData object. The query operator returns a derived SpatialData object that contains the data within the query region for all layers, including the corresponding table annotations. The bounding box spatial query can be performed in 2D for all elements or in 3D for raster elements (that is, Image and Labels) and points; extended discussion on 3D queries is presented in Supplementary Note 6. Shown are code excerpts from the spatial query tutorial. This specific tutorial explains how a region of interest can be specified, such as rectangular bounding boxes or defined via polygonal shapes, and how the data underlying the specified query region can be retrieved. The full example can be found in the 'spatial query' notebook in the online documentation (https://spatialdata.scverse.org/en/latest/tutorials/notebooks/notebooks/examples/spatial_query.html).

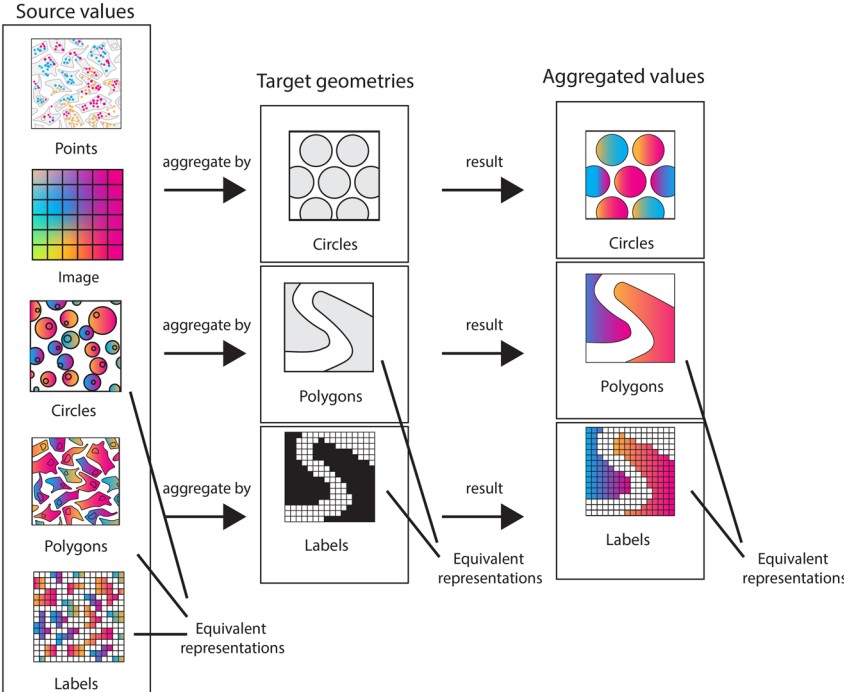

**Extended Data Fig. 2 | Schematic representation of the SpatialData aggregation operations.** Aggregation operations are the foundation to flexibly transfer quantifications and annotations across modalities when conducting multimodal analyses. SpatialData enables the aggregation (also referred to as accumulation in image processing) of data stored in any SpatialElement into any set of target geometries or masks. Example applications include count aggregation of the number of single molecules for a specific gene within polygon geometries representing cells. Similarly, molecule counts within image masks representing the cytoplasm of the cells. Another example is averaging cell gene expression within a given anatomical region (see also main text Fig. 2). Predefined aggregation operators (count, sum, mean, standard deviation) can be applied to any SpatialElement. Additionally, SpatialData supports the definition of user-specified aggregation operators. Leveraging common coordinate systems, aggregation operations can be applied to collections of datasets, including across datasets with different spatial scales and/or partially overlapping datasets. Tutorials on how to use the aggregation system are available as part of the SpatialData online documentation (https://spatialdata.scverse.org/en/latest/tutorials/notebooks/notebooks/examples/aggregation.html).

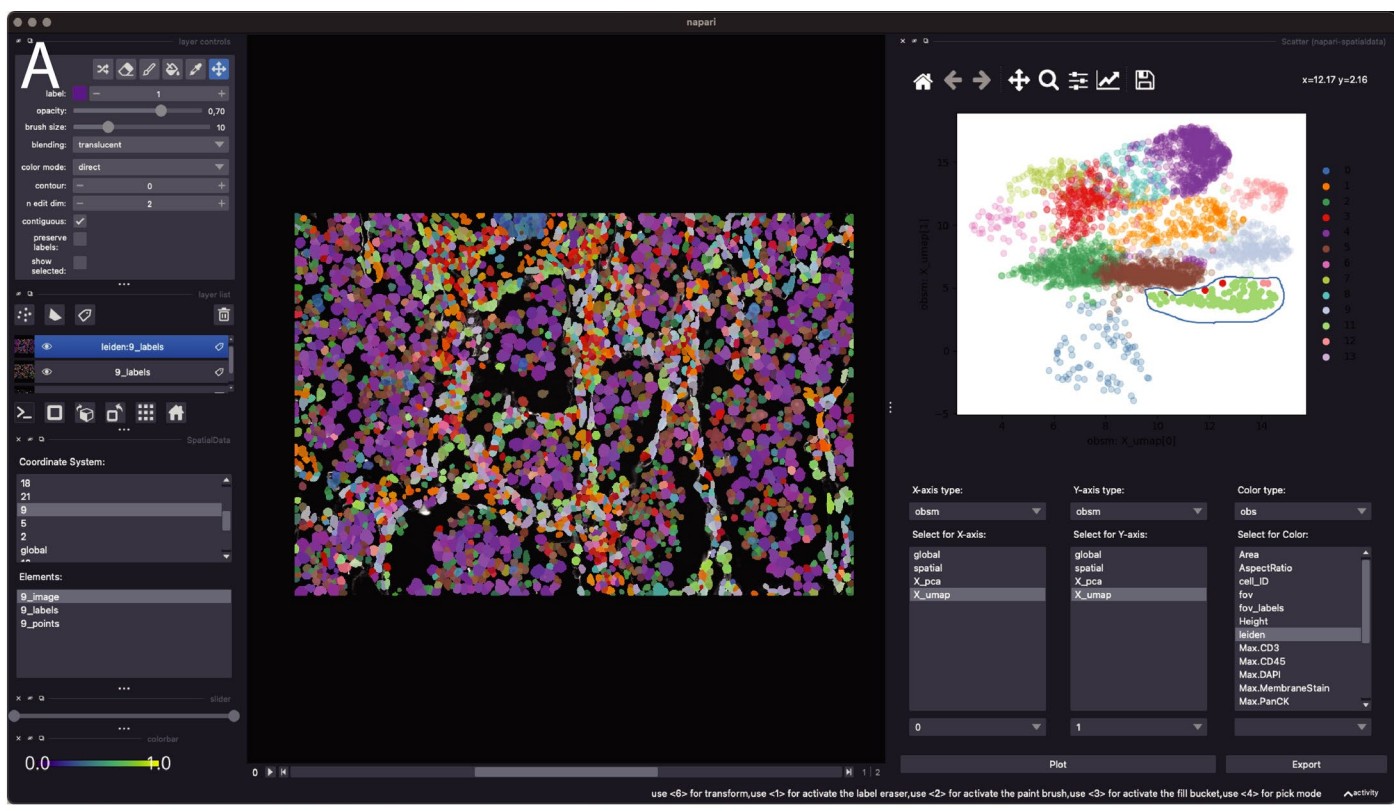

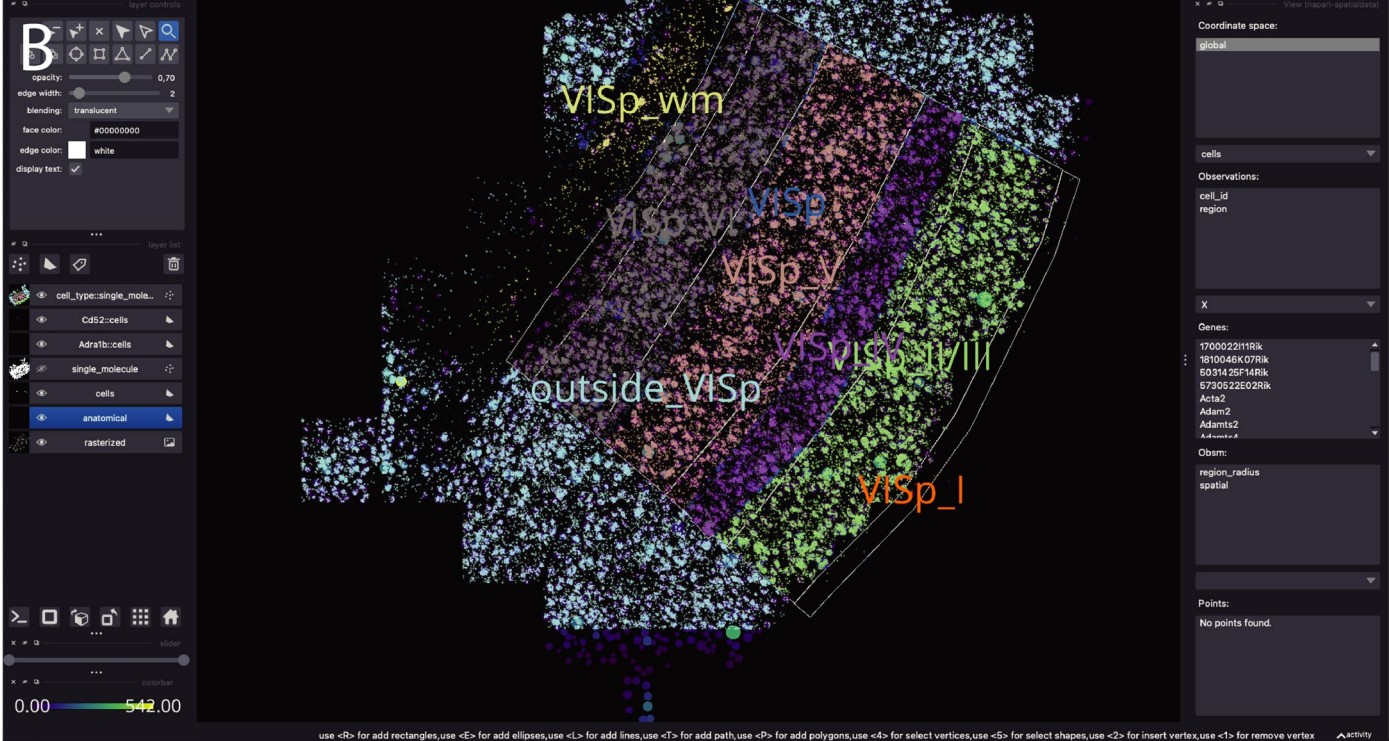

**Extended Data Fig. 3 | Example of using napari-spatialdata to visualize and annotate spatial datasets.** Napari-spatialdata enables the interactive visualization of SpatialElements (Images, Labels, Points, Shapes) together with associated annotations (such as gene expression, cluster annotations etc.). Embeddings of molecular profiles (for example, t-SNE, UMAP) can be interactively queried via the scatter plot widget. Spatial annotations can be interactively created via drawing of regions in the napari viewer. The corresponding annotations are then exported into the underlying SpatialData, facilitating their use in downstream analyses. **a**. NanoString CosMx dataset and interactive selection with a lasso from the UMAP plot computed from the cell gene expression and colored by Leiden clusters. The lasso tool in the scatterplot windows is used to annotate a set of cells. The annotation can be visualized in space and can be exported for downstream usage. **b**. MERFISH mouse brain dataset (Allen Institute prototype MERFISH pipeline[35]) featuring gene expression, polygonal ROIs annotating anatomical regions and cell types assigned to single molecule points.

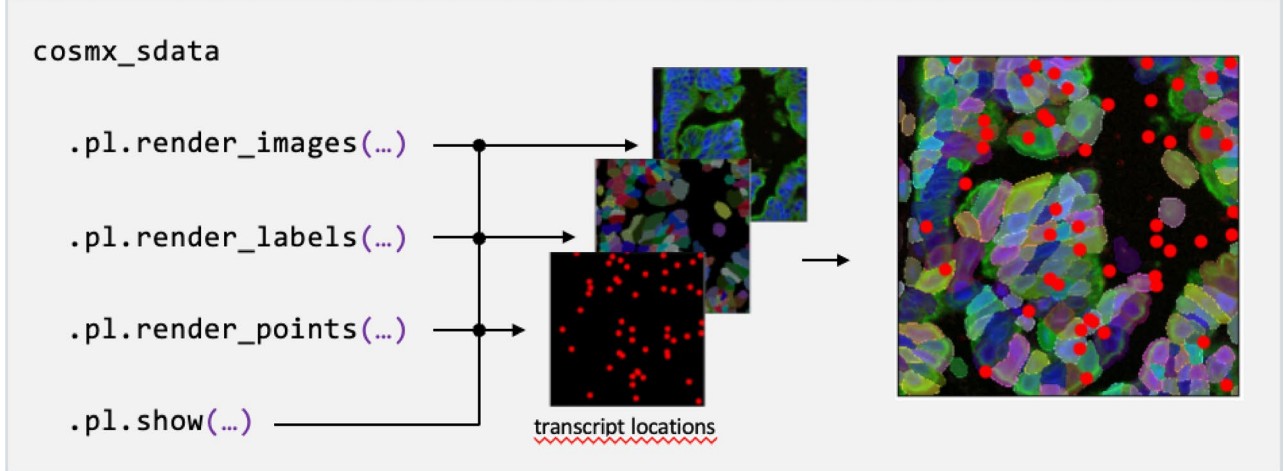

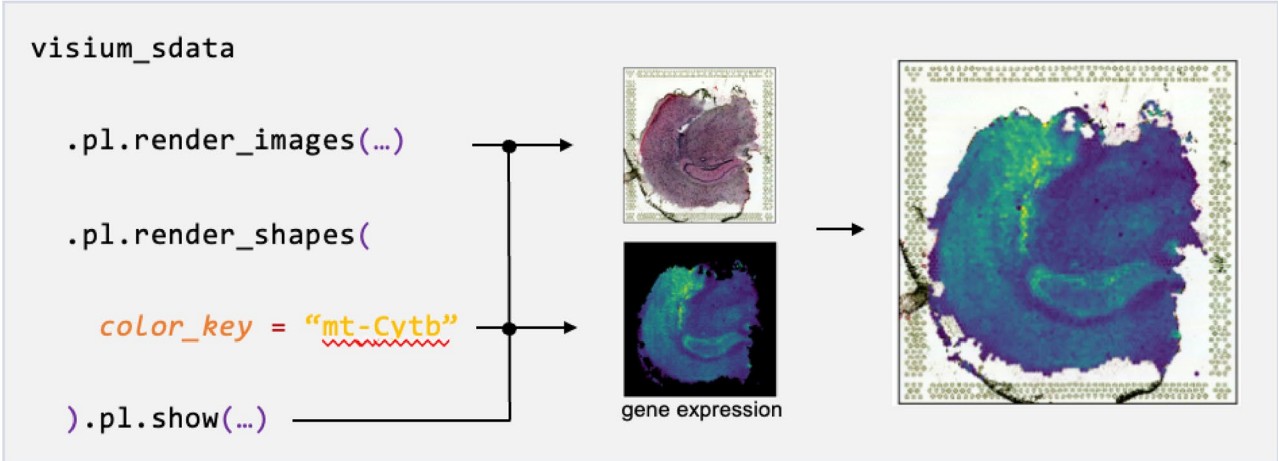

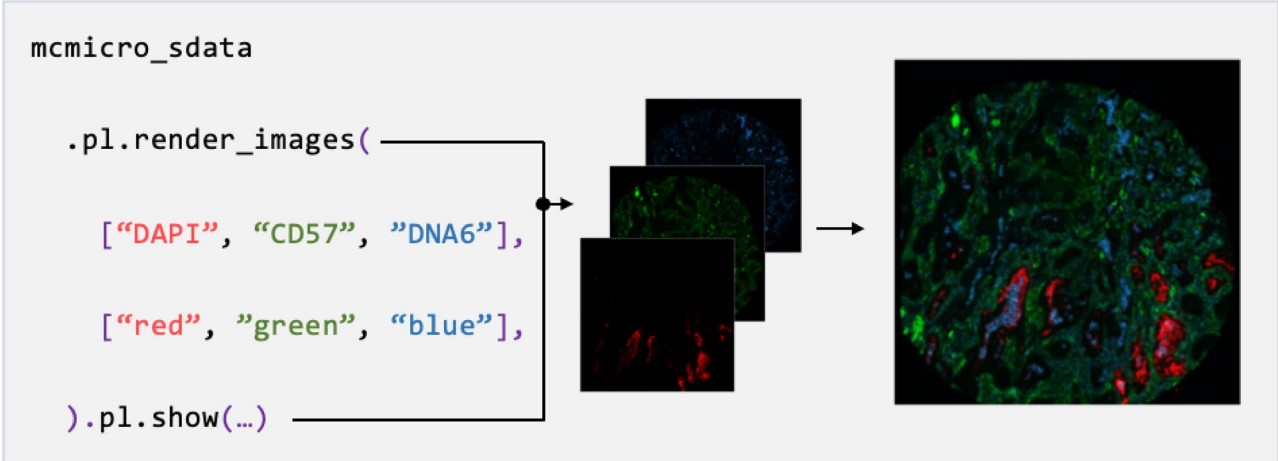

**Extended Data Fig. 4 | See next page for caption.**

**Extended Data Fig. 4 | Illustration of the static plotting library spatialdata-plot.** The spatialdata-plot library enables the streamlined visualization of complex multi-modality data. The set of elements to be rendered (Images, Labels, Points, Shapes), as well as specific parameters for plotted elements can be specified by the user. For example, Shapes representing cells can be annotated by the expression level of a target gene. The plotting library automatically accounts for transformations and alignments of the underlying common coordinate system. Tutorials how to use spatialdata-plot are available as part of the online documentation (Section 'Visualizations', https://spatialdata.scverse.org/en/latest/tutorials/notebooks/notebooks.html).

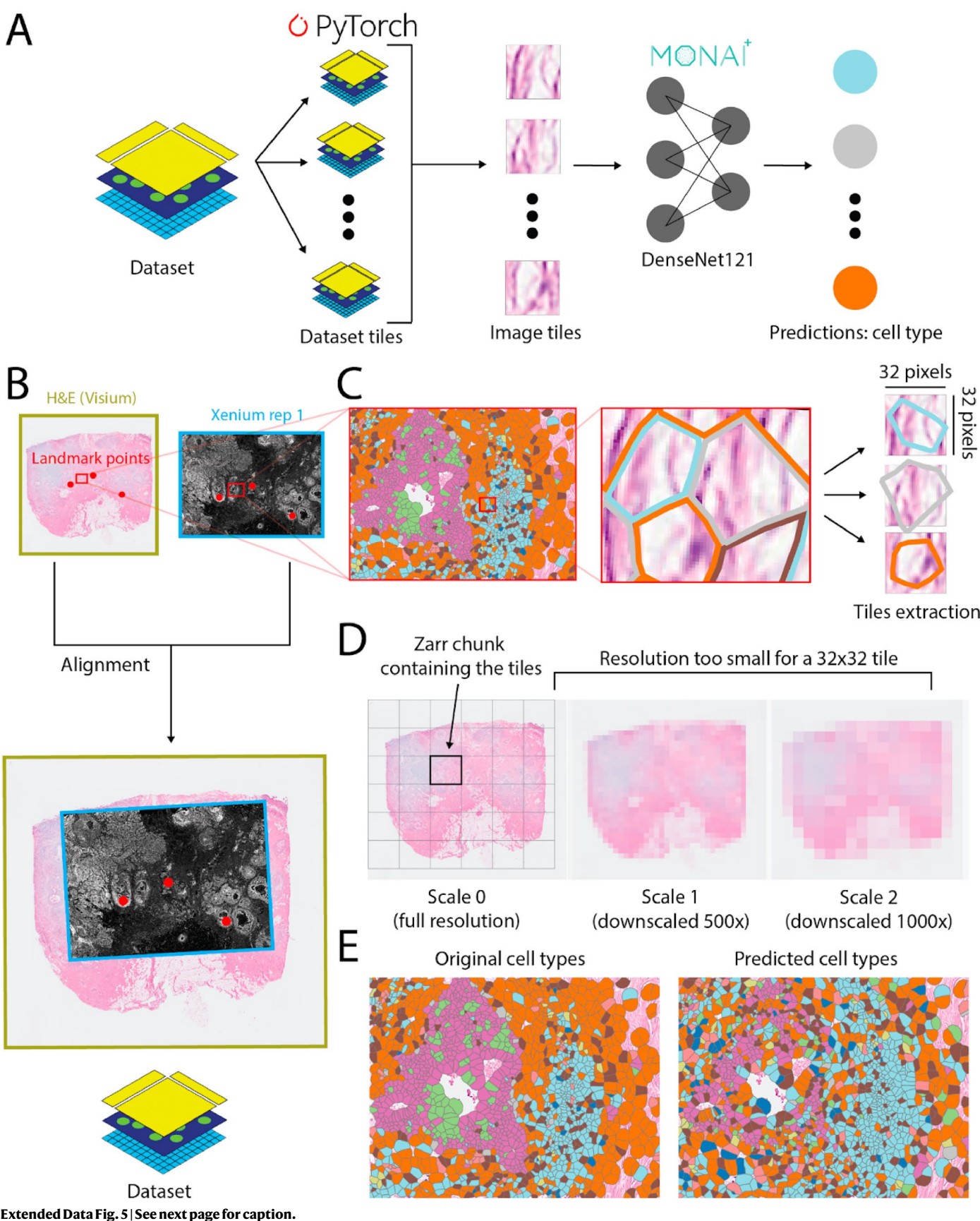

**Extended Data Fig. 5 | See next page for caption.**

**Extended Data Fig. 5 | SpatialData facilitates the preparation of datasets for deep learning applications and it integrates with existing deep learning ecosystems. (a)** Building on the query interface, SpatialData allows to generate PyTorch datasets that represent tiles of the original SpatialData. Shown is an example use case, using tiles centered on cells to train a DenseNet encoder model for supervised cell-type prediction. The specific model architecture, without weights, is provided by the MONAI framework, and this example shows how we can readily interface with existing deep learning ecosystems. **(b)** The effective definition of deep learning datasets can harness common coordinate systems to allow for the combination of different spatially aligned elements. Shown are H&E image and Xenium replicate 1 aligned datasets precedently introduced in main text Fig. 2a. **(c)** Enlarged view of a subset of the two datasets, overlaying the cells from Xenium, colored by cell type, to the H&E image from Xenium. SpatialData allows to extract image tiles of the desired resolution (here 32x32 pixels) around the Xenium cells. **(d)** The tiling extraction process takes advantage of the multiscale representation and the chunked Zarr storage for efficient memory usage. The first allows the extraction of the tiles from the appropriate (downscaled) resolution, the second ensures that only the data chunk(s) containing the information about the tiles are loaded from disk. Note: the 500x and 1000x downscaling factors and the size of the chunks have been chosen for illustrative purposes. **(e)** Visualization of cell-type labels predicted by the model. Note: due to the illustrative purpose of this example, focusing on the demonstration of the infrastructure, network training has been limited to a small number of epochs, and systematic hyperparameter optimization has been omitted. This is reflected in the suboptimal accuracy of the predictions. The full example can be found in the online documentation (https://spatialdata.scverse.org/en/latest/tutorials/notebooks/notebooks/examples/densenet.html).

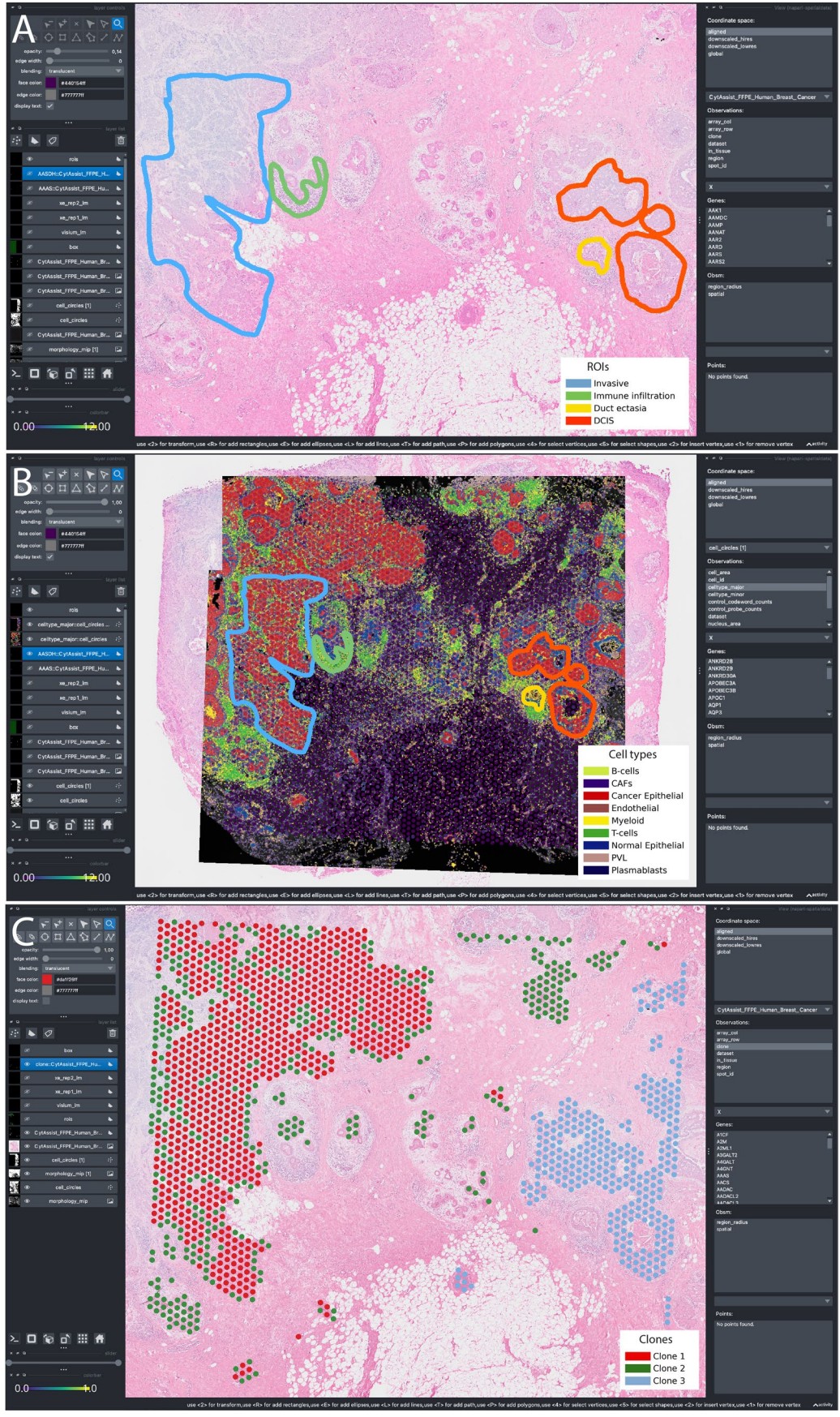

**Extended Data Fig. 6 | See next page for caption.**

**Extended Data Fig. 6 | Napari-based visualization of the Visium and the two Xenium datasets from the breast cancer study presented in main text. (a)** H&E image from the Visium dataset annotated with the ROIs for anatomically relevant tissue compartments. **(b)** Multimodal visualization of the H&E image from the Visium data, the two immunofluorescence images associated with the Xenium data, the Visium array capture locations colored by gene expression (showed with transparency), the Xenium cells showing cell types and the four manually annotated ROIs. **(c)** Visualization of the clone annotations estimated from Visium count data. Dedicated tutorials on how to use napari-spatialdata to align different modalities via landmark-based annotation and how to manually draw regions of interest, can be found in the online documentation (https://spatialdata.scverse.org/en/latest/tutorials/notebooks/notebooks/examples/alignment_using_landmarks.html, https://spatialdata.scverse.org/en/latest/tutorials/notebooks/notebooks/examples/napari_rois.html).

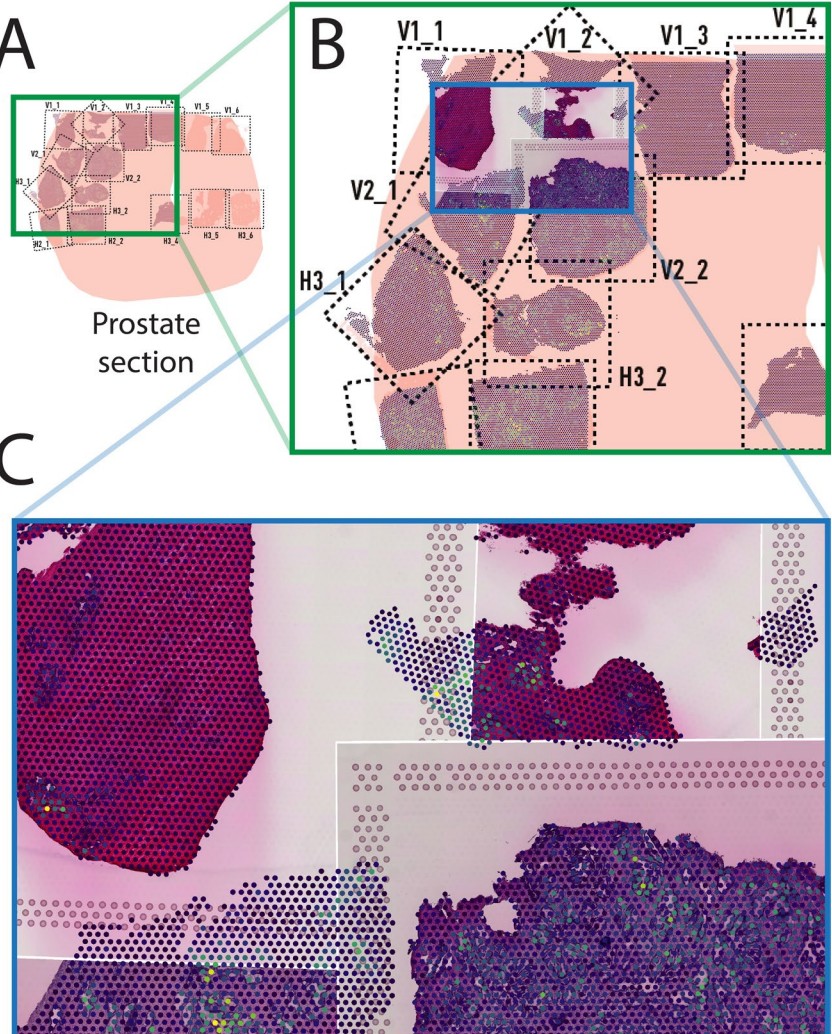

**Extended Data Fig. 7 | Example of using SpatialData to combine multiple datasets from a prostate cancer study into a common coordinate system.** Shown is a common coordinate system constructed using data from Erikson et al.[36]. The study comprises multiple Visium H&E and Spatial Transcriptomics[36] datasets from multiple tissue samples, with partially overlapping fields-of-view distributed across the tissues. **(a)** Spatial layout of the 15 fields-of-view for the Visium experiments for one of the tissues. Coordinate transformations used to align the fields-of-view were derived using SpatialData (landmark-based alignment), by aligning each image to the global layout image available from the original publication. **(b)** Screenshot of the visualization of all Visium datasets for one of the tissue samples in the context of the whole tissue coordinate system using napari-spatialdata. **(c)** The SpatialData multiscale image representation, napari-spatialdata allows to view and interactively explore all of the large images (15 images, ≈ 580 megapixels each) aligned together with the spatial gene expression. We can also visualize multiple modalities together, such as adding to the view also the Spatial Transcriptomics data. The full example can be found in (https://github.com/scverse/spatialdata-notebooks/blob/main/notebooks/paper_reproducibility/lundeberg.ipynb), and a dedicated tutorial on coordinate transformation can be found in the online documentation (https://spatialdata.scverse.org/en/latest/tutorials/notebooks/notebooks/examples/transformations.html). *The layout image used in the background in panels A and B is in the original publication[36] under the Creative Commons Attribution 4.0 International License. To view a copy of this license, visit* http://creativecommons.org/licenses/by/4.0/.

## Extended Data Table 1 | Comparison of alternative spatial omics analysis frameworks

| Method | Contribution | Access | Data Types | | | | | | | | | Operations | | | | | Plotting | |
|---|---|---|---|---|---|---|---|---|---|---|---|---|---|---|---|---|---|---|
| | | | Raster images | Raster labels | Multiscale raster | Polygons | Regular shapes | Points | Features matrix | Annotation matrix | Graphs | Points aggregation | Geometry intersection | Coordinate transformations | Coordinate systems | Interactive annotation | Static Plotting | Interactive Plotting |
| Voyager (SpatialFeatureExperiment) | framework | R, Python | Yes | Yes | No | Yes | Yes | Yes | Yes | Yes | Yes | Yes | Yes | No | No | No | Yes | No |
| SpatialExperiment | framework | R | Yes | No | No | No | Yes | Yes | Yes | Yes | Yes | No | No | No | No | No | Yes | No |
| Giotto | framework | R | Yes | Yes | No | No | Yes | Yes | Yes | Yes | Yes | Partial | No | No | No | Yes | Yes | Yes |
| MoleculeExperiment | framework | R | No | No | No | Yes | Yes | Yes | Yes | Yes | No | Yes | No | No | No | No | Yes | No |
| SODB | database | web, Python | Yes | Yes | No | No | Yes | No | Yes | Yes | Yes | No | Partial | No | No | No | Yes | Yes |
| STOmicsDB | database | web | Yes | No | No | No | No | No | Yes | Yes | Yes | No | No | No | No | No | Yes | Yes |
| emObject | framework | Python | Yes | Yes | No | No | No | No | Yes | Yes | No | No | No | Yes | Yes | No | Yes | No |
| Squidpy | framework | Python | Yes | Yes | No | No | Yes | No | Yes | Yes | Yes | No | No | No | No | Yes | Yes | Yes |
| SpatialData | framework | Python | Yes | Yes | Yes | Yes | Yes | Yes | Yes | Yes | Yes | Yes | Yes | Yes | Yes | Yes | Yes | Yes |

Shown are existing frameworks (rows), classified by their primary target goal (analysis framework versus database), access mode as well as supported data types, operations and interactive visualization capabilities. The following frameworks are considered: Voyager[28] (SpatialFeatureExperiments), SpatialExperiment[29], Giotto[30], MoleculeExperiment[31], SODB[32], STOmicsDB[33], emObject[34], Squidpy[15]. The Giotto 'Points aggregation' is classified as partial because the current implementation is limited to a regular grid as target geometry for aggregation. The SODB 'geometry interaction' is classified as partial as it is accessible via the web interface only, but not in programmatic fashion.

# Reporting Summary

## Statistics

For all statistical analyses, confirm that the following items are present in the figure legend, table legend, main text, or Methods section.

| n/a | Confirmed | |
|---|---|---|
| ☒ | ☐ | The exact sample size (*n*) for each experimental group/condition, given as a discrete number and unit of measurement |
| ☒ | ☐ | A statement on whether measurements were taken from distinct samples or whether the same sample was measured repeatedly |
| ☒ | ☐ | The statistical test(s) used AND whether they are one- or two-sided<br>*Only common tests should be described solely by name; describe more complex techniques in the Methods section.* |
| ☒ | ☐ | A description of all covariates tested |
| ☒ | ☐ | A description of any assumptions or corrections, such as tests of normality and adjustment for multiple comparisons |
| ☒ | ☐ | A full description of the statistical parameters including central tendency (e.g. means) or other basic estimates (e.g. regression coefficient) AND variation (e.g. standard deviation) or associated estimates of uncertainty (e.g. confidence intervals) |
| ☒ | ☐ | For null hypothesis testing, the test statistic (e.g. $F$, $t$, $r$) with confidence intervals, effect sizes, degrees of freedom and $P$ value noted<br>*Give P values as exact values whenever suitable.* |
| ☒ | ☐ | For Bayesian analysis, information on the choice of priors and Markov chain Monte Carlo settings |
| ☒ | ☐ | For hierarchical and complex designs, identification of the appropriate level for tests and full reporting of outcomes |
| ☐ | ☒ | Estimates of effect sizes (e.g. Cohen's *d*, Pearson's *r*), indicating how they were calculated |

*Our web collection on statistics for biologists contains articles on many of the points above.*

## Software and code

Policy information about availability of computer code

| | |
|---|---|
| Data collection | The data used in this study was downloaded from public sources using custom Python script that we made available at https://github.com/giovp/spatialdata-sandbox. Such scripts download the raw data and convert it to the SpatialData format. The converted data is also accessible (see Data Availability statement). |
| Data analysis | All code used to perform the analyses and generate figures in this manuscript is available here: https://github.com/scverse/spatialdata-notebooks/tree/main/notebooks/paper_reproducibility and has the following software requirements:<br>spatialdata>=0.0.15<br>spatialdata-io>=0.0.9<br>spatialdata-plot>=0.0.6<br>napari-spatialdata>=0.3.1<br>cell2location>=0.1.3<br>copykat>=1.1.0<br><br>The packages of the SpatialData framework (spatialdata, spatialdata-io, spatialdata-plot, napari-spatialdata) and all their dependencies can be installed automatically via pip, which is the recommended way to install the libraries (conda support is in preparation).<br>Here below is the list of dependent libraries and corresponding versions used at the time of writing this manuscript.<br>Requirements of the spatialdata package:<br>anndata>=0.9.1<br>numpy>=1.24.3<br>xarray>=2022.12.0<br>zarr>=2.14.2 |

```
ome_zarr>=0.7.0
spatial_image>=0.3.0
multiscale_spatial_image>=0.11.2
xarray-schema>=0.0.3
geopandas>=0.13.0
shapely>=2.0.1
rich>=13.3.1
pyarrow>=11.0.0
typing_extensions>=4.9.0
dask-image>=2022.9.0
networkx>=2.8.4
xarray-spatial>=0.3.5
tqdm>=4.65.0
Requirements of the spatialdata-io package:
scikit-image>=0.22.0
h5py>=3.9.0
imagecodecs>=2023.9.4
joblib>=1.3.2
readfcs>=1.1.7
Requirements of the spatialdata-plot package:
matplotlib>=3.8.2
scikit-learn>=1.3.2
scanpy>=1.9.6
matplotlib-scalebar>=0.8.1
Requirements of the napari-spatialdata package:
click>=8.1.7
cycler>=0.12.1
loguru>=0.7.2
napari>=0.4.16
napari-matplotlib>=1.2.0
numba>=0.58.1
packaging>=23.2
pillow>=10.0.0
qtpy>=2.4.1
scipy>=1.11.4
superqt>=0.6.1
vispy>=0.10.0
```

For manuscripts utilizing custom algorithms or software that are central to the research but not yet described in published literature, software must be made available to editors and reviewers. We strongly encourage code deposition in a community repository (e.g. GitHub). See the Nature Portfolio guidelines for submitting code & software for further information.

## Data

Policy information about availability of data

All manuscripts must include a data availability statement. This statement should provide the following information, where applicable:

- Accession codes, unique identifiers, or web links for publicly available datasets
- A description of any restrictions on data availability
- For clinical datasets or third party data, please ensure that the statement adheres to our policy

We converted several example datasets to Zarr using the SpatialData package (see Software and Code). At the time of writing, we include data from the following technologies: NanoString CosMx, 10x Genomics Xenium, 10x Genomics Visium, CyCIF, MERFISH, MIBI-TOF, Imaging Mass Cytometry. The converted data is accessible from https://spatialdata.scverse.org/en/latest/tutorials/notebooks/datasets/README.html. For an overview of the datasets and their respective source publication, please refer to Table S4.

## Human research participants

Policy information about studies involving human research participants and Sex and Gender in Research.

| | |
|---|---|
| Reporting on sex and gender | No participants have been recruited for this study. |
| Population characteristics | No participants have been recruited for this study. |
| Recruitment | No participants have been recruited for this study. |
| Ethics oversight | No participants have been recruited for this study. |

Note that full information on the approval of the study protocol must also be provided in the manuscript.

# Field-specific reporting

Please select the one below that is the best fit for your research. If you are not sure, read the appropriate sections before making your selection.

☒ Life sciences          ☐ Behavioural & social sciences          ☐ Ecological, evolutionary & environmental sciences

For a reference copy of the document with all sections, see nature.com/documents/nr-reporting-summary-flat.pdf

# Life sciences study design

All studies must disclose on these points even when the disclosure is negative.

| | |
|---|---|
| Sample size | NanoString CosMx data: 1 sample, 30 slides.<br>10x Genomics Xenium + Visium (breast cancer, Janesick et al.): 1 sample, 3 slides<br>CyCIF MCMICRO: 1 sample, 1 slide<br>MERFISH: 1 sample, 1 slide<br>MIBI-TOF: 3 samples, 1 slide each<br>Imaging Mass Cytometry: 4 patients, 14 slides total<br>10x Genomics Visium (prostate cancer, Erikson et al.): 1 sample, 15 fields-of-view<br><br>The sample sizes and slides number were chosen to be often greater than 1 to demonstrate the integration capabilities of the framework. |
| Data exclusions | No data has been excluded during our data processing. |
| Replication | The code to generate all figures is publicly available. See Software and Data Availability section. |
| Randomization | We did not require randomization in the data. We did not divide the data into subgroups. |
| Blinding | We did not divide the data into subgroups. |

# Reporting for specific materials, systems and methods

We require information from authors about some types of materials, experimental systems and methods used in many studies. Here, indicate whether each material, system or method listed is relevant to your study. If you are not sure if a list item applies to your research, read the appropriate section before selecting a response.

## Materials & experimental systems

| n/a | Involved in the study |
|---|---|
| ☒ ☐ | Antibodies |
| ☒ ☐ | Eukaryotic cell lines |
| ☒ ☐ | Palaeontology and archaeology |
| ☒ ☐ | Animals and other organisms |
| ☒ ☐ | Clinical data |
| ☒ ☐ | Dual use research of concern |

## Methods

| n/a | Involved in the study |
|---|---|
| ☒ ☐ | ChIP-seq |
| ☒ ☐ | Flow cytometry |
| ☒ ☐ | MRI-based neuroimaging |

