## [Peer Review File · Nature Methods]

Peer Review Information

Manuscript Title: SpatialData: an open and universal data framework for spatial omics

Corresponding author name(s): Josh Moore, Fabian Theis, Oliver Stegle

Editorial Notes:

Reviewer Comments & Decisions:

Decision Letter, initial version:
--

Dear Oliver,

Please let me begin by offering my sincerest apologies about the slow speed of your review (referee 2 also offered apologies).

Your Brief Communication, "SpatialData: an open and universal data framework for spatial omics", has now been seen by two reviewers. As you will see from their comments below, although the reviewers find your work of considerable potential interest, they have raised a small number of concerns. We are interested in the possibility of publishing your paper in Nature Methods, but would like to consider your response to these concerns before we reach a final decision on publication. We therefore invite you to revise your manuscript to address these concerns.

We thought all the suggestions were constructive, but we do not require you to add an interface to propose napari solutions (nice, but out of the scope of this paper) or make a 3D-compatible version, though this should be discussed. We hope the other concerns should be straightforward to address or clarify.

* include a point-by-point response to the reviewers and to any editorial suggestions

* please underline/highlight any additions to the text or areas with other significant changes to facilitate review of the revised manuscript

- * address the points listed described below to conform to our open science requirements
- * ensure it complies with our general format requirements as set out in our guide to authors at www.nature.com/naturemethods
- * resubmit all the necessary files electronically by using the link below to access your home page

[Redacted] This URL links to your confidential home page and associated information about manuscripts you may have submitted, or that you are reviewing for us. If you wish to forward this email to co-authors, please delete the link to your homepage.

We hope to receive your revised paper within two months. If you cannot send it within this time, please let us know. In this event, we will still be happy to reconsider your paper at a later date so long as nothing similar has been accepted for publication at Nature Methods or published elsewhere.

OPEN SCIENCE REQUIREMENTS

REPORTING SUMMARY AND EDITORIAL POLICY CHECKLISTS

DATA AVAILABILITY

We strongly encourage you to deposit all new data associated with the paper in a persistent repository where they can be freely and enduringly accessed. We recommend submitting the data to discipline-specific and community-recognized repositories; a list of repositories is provided here:

<http://www.nature.com/sdata/policies/repositories>

All novel DNA and RNA sequencing data, protein sequences, genetic polymorphisms, linked genotype and phenotype data, gene expression data, macromolecular structures, and proteomics data must be deposited in a publicly accessible database, and accession codes and associated hyperlinks must be provided in the “Data Availability” section.

Please include a “Data availability” subsection in the Online Methods. This section should inform readers about the availability of the data used to support the conclusions of your study, including accession codes to public repositories, references to source data that may be published alongside the paper, unique identifiers such as URLs to data repository entries, or data set DOIs, and any other statement about data availability. At a minimum, you should include the following statement: “The data that support the findings of this study are available from the corresponding author upon request”, describing which data is available upon request and mentioning any restrictions on availability. If DOIs are provided, please include these in the Reference list (authors, title, publisher (repository name), identifier, year). For more guidance on how to write this section please see: <http://www.nature.com/authors/policies/data/data-availability-statements-data-citations.pdf>

CODE AVAILABILITY

Please include a “Code Availability” subsection in the Online Methods which details how your custom code is made available. Only in rare cases (where code is not central to the main conclusions of the paper) is the statement “available upon request” allowed (and reasons should be specified).

For more information on our code sharing policy and requirements, please see: <https://www.nature.com/nature-research/editorial-policies/reporting-standards#availability-of-computer-code>

MATERIALS AVAILABILITY

ORCID

Sincerely,
Rita

Rita Strack, Ph.D.
Senior Editor
Nature Methods

Reviewers' Comments:

Reviewer #2:
Remarks to the Author:

The SpatialData framework introduced in the well-written manuscript by Marconato et al. describes a comprehensive solution for an urgent need in the spatial omics domain. It will make data analysis and tool building more efficient and also provides a conceptual framework to think about integration of spatial data. SpatialData builds on existing emerging standards such as OME-NGFF and in extension, Zarr, which simplifies adoption and integration with existing tools and frameworks.

Important questions that I still have after reviewing the manuscript:

1. If SpatialData is designed to support 3D spatial data with a z-dimension and if so, what kind of queries are supported. If there is no support for 3D data at the moment, could this be added in the

future?

2. What is the effective scalability of the format? What issues are to be expected when dozens or hundreds of image are included?
3. For an application in which all multi-modal data for a given region needs to be retrieved, is there support for a query that returns all "layers" for a given x,y window?
4. How would one access the data if the Python library is not an option, e.g., directly from JavaScript in a web browser for visualization purposes or from R/Bioconductor code?
5. In general, how does SpatialData relate to similar efforts, e.g., SpatialFeatureExperiment in the Bioconductor project and how compatible is SpatialData with downstream analysis conducted outside of the Python single-cell ecosystem?

One possible improvement of the manuscript would be to describe an interface or schema for characterizing and including regions and landmarks and then propose the napari plugin as one possible solution for this task. In many cases, landmarks and other image regions of interest will be generated by other tools (e.g., vendor proprietary software such as the Nanostring GeoMX tool suite) and then has to be converted into the appropriate format for inclusion in SpatialData.

A very minor point is that the authors switch between "common coordinate system (CCS)" and "common coordinate framework (CCF)" in the text. I assume that mean the same and they should probably stick one term (CCF being the more popular one).

Reviewer #3:

Remarks to the Author:

Summary

This paper describes a new framework for the unified storage and handling of spatial transcriptomics (ST) data. The new SpatialData object is uniquely tailored to harmonize disparate methods for data production, aggregate measurements across modalities, and align spatial data through the use of waypoints. Further, the object appeals to both sequencing and imaging communities through the extension of existing single cell packages (e.g. SquidPy) and through the use of the Zarr OME-NGFF specification. In the manuscript and accompanying documents, the authors provide a complete description of the technology through a set of tutorials, and supplementary notes. Finally, the development of the napari-spatialdata plugin allows for easy visualization of the new data in a locally hosted 3D viewer.

Major Points

No major or minor concerns. The manuscript's text, figures and supplementary notes are well written and presented.

Author Rebuttal to Initial comments
--

Point-by-point response to reviewer comments on NMETH-BC52589

Reviewers' comments are *italicized*; author responses are presented directly after each comment. Text added in the latest manuscript submission is highlighted in green.

R#2 preamble: *"The SpatialData framework introduced in the well-written manuscript by Marconato et al. describes a comprehensive solution for an urgent need in the spatial omics domain. It will make data analysis and tool building more efficient and also provides a conceptual framework to think about integration of spatial data..."*

We thank the reviewer for the positive comments and for recognizing the community need for a framework such as SpatialData.

R#2C1: *"If SpatialData is designed to support 3D spatial data with a z-dimension and if so, what kind of queries are supported. If there is no support for 3D data at the moment, could this be added in the future?"*

We agree that the support for emerging 3D spatial datasets is an interesting direction to consider. While 3D is not the focus of the current release of SpatialData, the framework is intrinsically designed to support 3D operations to cater to future needs. Specifically, we have already implemented 3D support to store such data, as well as query operations on the Image and Labels elements, thereby setting the foundation for future developments in this direction. Having said this, we feel that comprehensive 3D support is out of scope for this manuscript, primarily because the corresponding datasets are still extremely scarce. Thus, in line with the editorial recommendations, we have clarified the current level of 3D support in SpatialData and we have highlighted full 3D support as an area of future development.

Revised text in Supplementary Note 3, clarifying that 3D queries are supported for selected elements

"When operating on large data, it can be helpful to subdivide the dataset into multiple regions. The spatial query interface allows users to request the data contained in a query region, which can be specified both as a bounding box or a polygonal region. The query region can be defined in any coordinate system present in the SpatialData object. The result of this query operation is returned as a derived SpatialData object, containing the data within the query region for all layers. The spatial query functionality is illustrated in interactive tutorials that are part of the SpatialData online documentation (Supplementary Figure 2). *The bounding box spatial query can be performed in 2D for all elements or 3D for raster elements (i.e., Image and Labels). Shapes cannot currently be queried or aggregated in 3D as the current implementation treats them as 2D polygons (in line with current commercial technologies). Full 3D support in SpatialData will require the ability to store 3D*

vector objects (e.g., polygonal meshes). The corresponding software extensions are an area of future work.”

Revised text in Discussion, describing the opportunity to further develop full 3D support

“Ongoing development will extend the interoperability of SpatialData with R/Bioconductor¹ (Supplementary Note 13), provide support for multiscale point and polygon representations, increase support for 3D data types and operations, such as polygonal meshes as well as 5D volumetric images (i.e., *czyx* images with an additional time component), and support cloud-based data access both programmatically and via the visualization tool Vitessece². In summary, SpatialData fills the important need for an open and universal data framework for spatial omics.”

R#2C2: *“What is the effective scalability of the format? What issues are to be expected when dozens or hundreds of images are included?”*

We thank the reviewer for pointing out this important consideration. We have designed the SpatialData storage format specifically with scalability of large image data in mind. Specifically, the SpatialData storage format builds on top of the OME-Zarr format, which has previously been demonstrated to scale to datasets with up to hundreds of terabytes of size (Moore et al., *Histochemistry and Cell Biology*, 2023). The primary limitation in scaling to large images is the need for an appropriate data storage backend that can handle both large data volumes and a large number of files. We have added a supplementary note (Supplementary Note 12), discussing the scalability and constraints when storing large images in the SpatialData storage format.

New supplementary note 12, describing scalability and considerations for storing large images in SpatialData

Supplementary Note 12: Scalability of image storage in SpatialData

To ensure scalability to large datasets, SpatialData relies on the next-generation file format, OME-Zarr, which has been specifically designed for processing and visualization of large scientific data^{3,4}. The raw image data tends to comprise the largest portion of a (processed) spatial omics dataset, thus is the key data storage bottleneck. As file sizes grow, OME-Zarr can be over 10 times faster than traditional image formats such as TIFF when accessing from local storage and over 100 times faster when accessing data from cloud storage³.

Briefly, OME-Zarr implements chunked storage of binary, compressed data to support performant parallel writing and reading of large image data. Consequently, OME-Zarr

scales to datasets of hundreds of terabytes in size ⁴, either stored as individual image volumes or collections of images. Additionally, OME-Zarr supports a multiscale representation of each image (i.e. image pyramid) to enable seamless zooming during interactive viewing of high resolution images and segmentation masks. Through this underlying infrastructure, SpatialData can load data lazily, permitting pipelines utilizing SpatialData to efficiently load and process large datasets.

We note that in practice the performance and scalability of processing large data will critically depend on the underlying data storage and computation infrastructure. In particular, the ability to lazily load data in parallel is closely tied to the chunk configuration of Zarr stores. In practice, this can lead to a large number of files per image. Object storage such as S3 is often the preferred file system for handling large file numbers. Additionally, the next major version of Zarr, v3, will support “sharding” which reduces the number of files by storing multiple chunks in a single file ⁵.

R#2C3: *“For an application in which all multi-modal data for a given region needs to be retrieved, is there support for a query that returns all “layers” for a given x,y window?”*

Yes, indeed, the SpatialData query interface supports this operation via the `bounding_box_query()` function. We have clarified the description in Supplementary Note 3 (revised text below) and updated the corresponding example notebook (https://spatialdata.scverse.org/en/latest/tutorials/notebooks/notebooks/examples/spatial_query.html), portrayed in Supplementary Figure 2, to highlight this feature.

New text in Supplementary Note 3

“When operating on large data, it can be helpful to subdivide the dataset into multiple regions. The spatial query interface allows users to request the data contained in a query region, which can be specified both as a bounding box or a polygonal region. The query region can be defined in any coordinate system present in the SpatialData object. The result of this query operation is returned as a derived SpatialData object, containing the data within the query region for all layers. The spatial query functionality is illustrated in an interactive tutorial that is part of the SpatialData online documentation. Figure S2 shows fragments of the code from the documentation, and the resulting query data overlaid on the original dataset.

Supplementary Figure 2 | Illustration of the SpatialData query function. Spatial queries allow for retrieving data within a specified spatial region (e.g., a bounding box or a polygon). Shown are fragments of the code from the spatial query tutorial, which is part of the online SpatialData documentation. The tutorial explains how a region of interest can be specified, such as rectangular bounding boxes or defined via polygonal shapes, and how to use them to retrieve all the layers underlying the specified query region. The full example can be found in the “spatial query” notebook in the online documentation (https://spatialdata.scverse.org/en/latest/tutorials/notebooks/notebooks/examples/spatial_query.html).

R#2C4: “How would one access the data if the Python library is not an option, e.g., directly from JavaScript in a web browser for visualization purposes or from R/Bioconductor code?”

We agree with the reviewer that interoperability is a key consideration for a framework such as SpatialData. SpatialData is designed to build on top of OME-NGFF, an open standard implemented in Zarr³. Zarr is an open format that can be readily accessed from different languages, including JavaScript⁶ and R⁷. Similarly, vector geometries of SpatialData objects are stored using Parquet, which can also be accessed from JavaScript and R. Thus, the data stored in SpatialData is in principle readily accessible from Python, JavaScript and R. Concerning the implementation of dedicated libraries to

provide seamless access, we strongly believe that such efforts have to be undertaken in collaboration with the respective communities. We have established such relationships and are in active exchange with the Vitesse² team and the Bioconductor community, to develop tailored SpatialData readers for JavaScript and R, respectively. To support these efforts, we provide example datasets (see Table S4), and we have created simple test datasets that provide examples of the core features of the format and also made them publicly available on the web. The new Supplementary Note 13 describes these new interoperability test datasets.

We also note that community efforts to provide dedicated access libraries from R and Javascript are currently underway:

- A team of Bioconductor contributors started working on an R in-memory and lazy SpatialData object, of which its preliminary capabilities can be seen here: <https://htmlpreview.github.io/?https://github.com/HelenaL.C/SpatialData/blob/development/vignettes/SpatialData.html>.
- Another team of Bioconductor contributors, which developed cytomapper⁸ and cytoviewer⁹, implemented preliminary support for OME-Zarr and the SpatialData storage format: <https://github.com/BodenmillerGroup/cytomapper/pull/82>.
- The Vitesse team is working towards interoperability with JavaScript; the current state of these efforts can be seen here: <https://github.com/vitesse/vitesse/issues/1292>.

We expect that these efforts will deliver fully-featured interfaces in due course. These community efforts highlight the possibility for SpatialData to facilitate interoperability across ecosystems.

New supplementary note describing the interoperability test datasets for methods developers

Supplementary Note 13: reference datasets for facilitating interoperability

To ensure SpatialData is interoperable with other spatial omics analysis communities, we have implemented the SpatialData format using storage technologies with broad reader support. In particular, we have targeted JavaScript as it is well-suited for web viewers and R because it is one of the most popular languages for bioinformatics. Since we have implemented the storage format in Zarr and Parquet, SpatialData files can be accessed in JavaScript and R using existing readers. For example, Zarr raster data can be read using: zarr.js⁶ (JavaScript) and Rarr⁷ (R), and Parquet vector geometries can be read using parquet.js (JavaScript) and arrow (R). Currently, tabular data stored in Zarr following the AnnData specification can be read in JavaScript (see <https://anndata.readthedocs.io/en/latest/interoperability.html>), and ongoing external work is being carried out to extend AnnDataR to support Zarr files (see <https://github.com/scverse/anndataR/issues/91>).

To make it seamless for users to load SpatialData files in JavaScript and R, we aim to support the work of the developers of spatial omics projects in those communities (e.g., Vitesse², SpatialFeatureExperiment¹⁰, SpatialExperiment¹¹) in developing specific readers for the SpatialData storage format. To support developers make readers in other

programming languages, we have created test datasets that provide examples the core parts of the specification and made them publicly available on the web (Table S5). Developers can use these test datasets to quickly verify that their readers are compliant with the SpatialData specification. Furthermore, developers can also test their methods with the example datasets from existing studies that we converted and made available in the cloud. The description of these datasets is given in the Table S4, and they can be accessed from the SpatialData online documentation.

Table S5 | Publicly available test datasets for verifying that readers are compliant with the SpatialData format specification. We are continuing to refine the test datasets as we receive feedback, to facilitate developers in reading data stored using the SpatialData storage format. For an up to date list, please consult the online documentation:

<https://spatialdata.scverse.org/en/latest/tutorials/notebooks/datasets/README.html>, under "Additional resources for methods developers".

Dataset	Scope
multiple_elements.zarr	Test a SpatialData object with multiple elements: 2D single-scale and multi-scale raster types (Image, Labels); 2D vector geometries (Points, Shapes: circle, polygons, multipolygons); Table annotating the 2D Labels.
transformation_identity.zarr	Test an Identity transformation on a 2D raster type (Image) and a 2D vector type (Points).
transformation_scale.zarr	Test a Scale transformation on a 2D raster type (Image) and a 2D vector type (Points).
transformation_translation.zarr	Test a Translation transformation on a 2D raster type (Image) and a 2D vector type (Points).
transformation_affine.zarr	Test an Affine transformation on a 2D raster type (Image) and a 2D vector type (Points).
transformation_sequence.zarr	Test a Sequence transformation on a 2D raster type (Image) and a 2D vector type (Points).

R#2C5: "In general, how does SpatialData relate to similar efforts, e.g., SpatialFeatureExperiment in the Bioconductor project and how compatible is SpatialData with downstream analysis conducted outside of the Python single-cell ecosystem?"

First of all, we note that SpatialData is designed to complement existing solutions and as outlined in the response to R#2C4, it is already interoperable with other spatial omics data management solutions and there are ongoing efforts to increase the interoperability. In response to this comment and R#2C4, we have detailed the strategy and current state to foster interoperability, including with the Bioconductor ecosystem. With regards to how SpatialData relates to SpatialFeatureExperiment in particular, we apologize for the lack of clarity in the previous submission. We have revised Table S1, which summarizes core features provided by SpatialData and alternative solutions. While SpatialFeatureExperiment provides many of the annotations and datatypes covered by SpatialData, there are important differences. A key distinction is stronger support for large images via a multiscale raster datatype offered by SpatialData, and that the framework comes with explicit support for aligning multiple datasets via common coordinate systems and transformations (Table S1). Table S2, which is now included in Supplementary Note 1, provides a comparison between the elements in SpatialData and the fields on the SpatialFeatureExperiment object, thus establishing a one-to-one mapping between the data types and modalities supported by both frameworks.

As outlined in response to comment R2C4, in terms of analysis outside of the Python single-cell ecosystem, examples of interoperable workflows enabled by the SpatialData storage format can be found among the preliminary efforts from external developers discussed previously. An instance of this is provided in the notebook (<https://htmlpreview.github.io/?https://github.com/HelenalC/SpatialData/blob/devel/vignettes/SpatialData.html>), which shows an example of how the gene expression AnnData table can be loaded into a SingleCellExperiment R object, how the Points can be loaded into a MoleculeExperiment R object, and how to use this data in downstream tasks such as filtering and plotting. In general, the development of seamless interfaces is an ongoing activity that is tackled in collaboration with the respective communities.

Updated Table S1 to clarify the relationship between Voyager and SpatialFeatureExperiment

Method	Contribution	Access	Data Types									Operations					Plotting	
			Raster images	Raster labels	Multiscale raster	Polygons	Regular shapes	Points	Features matrix	Annotation matrix	Graphs	Points aggregation	Geometry intersection	Coordinate transformations	Coordinate systems	Interactive annotation	Static Plotting	Interactive Plotting
Voyager (SpatialFeatureExperiment)	framework	R, Python	Yes	Yes	No	Yes	Yes	Yes	Yes	Yes	Yes	Yes	Yes	No	No	No	Yes	No
SpatialExperiment	framework	R	Yes	No	No	No	Yes	Yes	Yes	Yes	Yes	No	No	No	No	No	Yes	No
GeoTiff	framework	R	Yes	Yes	No	No	Yes	Yes	Yes	Yes	Yes	Partial	No	No	No	Yes	Yes	
MoleculeExperiment	framework	R	No	No	No	Yes	Yes	Yes	Yes	Yes	No	Yes	No	No	No	No	Yes	No
SCDB	database	web, Python	Yes	Yes	No	No	Yes	No	Yes	Yes	Yes	No	Partial	No	No	No	Yes	Yes
STORosDB	database	web	Yes	No	No	No	No	No	Yes	Yes	Yes	No	No	No	No	No	Yes	Yes
smObject	framework	Python	Yes	Yes	No	No	No	No	Yes	Yes	No	No	No	Yes	Yes	No	Yes	No
Squidpy	framework	Python	Yes	Yes	No	No	Yes	No	Yes	Yes	Yes	No	No	No	No	Yes	Yes	Yes
SpatialData	framework	Python	Yes	Yes	Yes	Yes	Yes	Yes	Yes	Yes	Yes	Yes	Yes	Yes	Yes	Yes	Yes	Yes

Table S2 | Comparison between SpatialData and SpatialFeatureExperiment on how elements that constitute a spatial omics dataset are stored and represented in memory.

Description	SpatialData	SpatialFeatureExperiment
Raster images	Image	ImgData
Raster multiscale images	Image	-
Raster labels (segmentation masks)	Labels	annotGeometries
Raster multiscale labels (segmentation masks)	Labels	-
Polygons	Shapes	colGeometries
Points (e.g. transcripts)	Points	rowGeometries
Graphs	Adata.obsp	colGraphs
Table annotations	Adata.X, Adata.var, Adata.obs	assays, rowData, colData

R#2C6: *“One possible improvement of the manuscript would be to describe an interface or schema for characterizing and including regions and landmarks and then propose the napari plugin as one possible solution for this task. In many cases, landmarks and other image regions of interest will be generated by other tools (e.g., vendor proprietary software such as the Nanostring GeoMX tool suite) and then have to be converted into the appropriate format for inclusion in SpatialData.”*

We thank the reviewer for the suggestion. We agree that it is important to be able to load regions/images from all available spatial omics instruments as easily as possible. After consultation with the editor, we believe making an additional interface available through the napari plugin is out of scope for this manuscript. Having said this, we note that all regions/images stored in the SpatialData storage format can be opened in the napari plugin. Detailed documentation describing the specification for the storage format is publicly available (https://spatialdata.scverse.org/en/latest/design_doc.html), and we encourage technology vendors to offer the option to export their region/landmarks in the format used by SpatialData. To help developers create writers compatible with SpatialData, we have added a new supplementary note describing resources for making tools and technologies compatible with SpatialData (Supplementary Note 13, described in response to **R#2C4**)

R#2C7: “A very minor point is that the authors switch between “common coordinate system (CCS)” and “common coordinate framework (CCF)” in the text. I assume that means the same and they should probably stick to one term (CCF being the more popular one).”

We appreciate the reviewer pointing out this inconsistency. Indeed, CCS and CCF were intended to refer to the same concept. We have unified the language and now use “common coordinate system (CCS)” throughout. We have opted for CCS as our understanding is that CCF refers to a coordinate system tied to anatomical landmarks. While the CCS are more general and can be used for any user-defined coordinate system (including CCFs)¹².

R#3: “No major or minor concerns. The manuscript’s text, figures and supplementary notes are well written and presented.”

We appreciate the reviewer noting that the work is well written and presented.

1. Gentleman, R. C. *et al.* Bioconductor: open software development for computational biology and bioinformatics. *Genome Biol.* **5**, R80 (2004).
2. Keller, M. S. *et al.* Vitessce: a framework for integrative visualization of multi-modal and spatially-resolved single-cell data. Preprint at <https://doi.org/10.31219/osf.io/y8thv> (2021).
3. Moore, J. *et al.* OME-NGFF: a next-generation file format for expanding bioimaging data-access strategies. *Nat. Methods* **18**, 1496–1498 (2021).
4. Moore, J. *et al.* OME-Zarr: a cloud-optimized bioimaging file format with international community support. *bioRxiv* 2023.02.17.528834 (2023) doi:10.1101/2023.02.17.528834.
5. Zarr core specification (version 3.0) — Zarr specs documentation. <https://zarr-specs.readthedocs.io/en/latest/v3/core/v3.0.html>.
6. Zuidhof, G. *zarr.js: Javascript implementation of Zarr.* (Github).
7. Rarr. *Bioconductor* <https://bioconductor.org/packages/release/bioc/html/Rarr.html>.
8. Eling, N., Damond, N., Hoch, T. & Bodenmiller, B. cytomapper: an R/Bioconductor package for visualization of highly multiplexed imaging data. *Bioinformatics* **36**, 5706–5708 (2021).
9. Meyer, L., Eling, N. & Bodenmiller, B. cytoviewer: an R/Bioconductor package for interactive visualization and exploration of highly multiplexed imaging data. *bioRxiv* (2023) doi:10.1101/2023.05.24.542115.

10. Moses, L., Jackson, K., Luebbert, L. & Pachter, L. Voyager: From geospatial to spatial omics. Preprint at <https://github.com/pachterlab/voyager> (2023).
11. Righelli, D. *et al.* SpatialExperiment: infrastructure for spatially resolved transcriptomics data in R using Bioconductor. *Cold Spring Harbor Laboratory* 2021.01.27.428431 (2021) doi:10.1101/2021.01.27.428431.
12. Rood, J. E. *et al.* Toward a Common Coordinate Framework for the Human Body. *Cell* **179**, 1455–1467 (2019).

Decision Letter, first revision:

Dear Oliver,

Thank you for submitting your revised manuscript "SpatialData: an open and universal data framework for spatial omics" (NMEM-BC52589A). It has now been seen by the original referees and their comments are below. The reviewers find that the paper has improved in revision, and therefore we'll be happy in principle to publish it in Nature Methods, pending minor revisions to comply with our editorial and formatting guidelines.

TRANSPARENT PEER REVIEW

ORCID

Sincerely,
Rita

Rita Strack, Ph.D.
Senior Editor

Nature Methods

Reviewer #2 (Remarks to the Author):

The authors fully addressed my questions and I am grateful for their clarifications in the revised manuscript and response.

Nils Gehlenborg

Reviewer #3 (Remarks to the Author):

No additional comments.

Final Decision Letter:

Dear Oli,

I am pleased to inform you that your Brief Communication, "SpatialData: an open and universal data framework for spatial omics", has now been accepted for publication in Nature Methods. The received and accepted dates will be May 15, 2023 and Feb 14, 2024. This note is intended to let you know what to expect from us over the next month or so, and to let you know where to address any further questions.

Over the next few weeks, your paper will be copyedited to ensure that it conforms to Nature Methods style. Once your paper is typeset, you will receive an email with a link to choose the appropriate publishing options for your paper and our Author Services team will be in touch regarding any additional information that may be required.

Once proofs are generated, they will be sent to you electronically and you will be asked to send a corrected version within 48 hours. It is extremely important that you let us know now whether you will be difficult to contact over the next month. If this is the case, we ask that you send us the contact information (email, phone and fax) of someone who will be able to check the proofs and deal with any last-minute problems.

If, when you receive your proof, you cannot meet the deadline, please inform us at rjsproduction@springernature.com immediately.

If you have posted a preprint on any preprint server, please ensure that the preprint details are updated with a publication reference, including the DOI and a URL to the published version of the

article on the journal website.

Please note that *Nature Methods* is a Transformative Journal (TJ). Authors may publish their research with us through the traditional subscription access route or make their paper immediately open access through payment of an article-processing charge (APC). Authors will not be required to make a final decision about access to their article until it has been accepted. Find out more about Transformative Journals

If you are active on Twitter/X, please e-mail me your and your coauthors' handles so that we may tag you when the paper is published.

To assist our authors in disseminating their research to the broader community, our SharedIt initiative provides you with a unique shareable link that will allow anyone (with or without a subscription) to read the published article. Recipients of the link with a subscription will also be able to download and print the PDF. As soon as your article is published, you will receive an automated email with your shareable link.

Please note that you and your coauthors may order reprints and single copies of the issue containing your article through Springer Nature Limited's reprint website, which is located at <http://www.nature.com/reprints/author-reprints.html>. If there are any questions about reprints please send an email to author-reprints@nature.com and someone will assist you.

Best regards,
Rita

Rita Strack, Ph.D.

Senior Editor
Nature Methods